# How well are aerosol-cloud interactions represented in climate models? Part 1: Understanding the sulphate aerosol production from the 2014-15 Holuhraun eruption.

George Jordan[1], Florent Malavelle[2], Ying Chen[3], Amy Peace[4], Eliza Duncan[4], Daniel G. Partridge[4], Paul Kim[4], Duncan Watson-Parris[5], Toshihiko Takemura[6], David Neubauer[7], Gunnar Myhre[8], Ragnhild Skeie[8], Anton Laakso[9], and James Haywood[1,4]

[1] Met Office Hadley Centre, Exeter, UK
[2] Met Office, Exeter, UK
[3] School of Geography Earth and Environment Sciences, University of Birmingham, UK
[4] College of Engineering, Mathematics, and Physical Sciences, University of Exeter, Exeter, UK
[5] Scripps Institution of Oceanography and Halicioğlu Data Science Institute, University of California San Diego, La Jolla, CA, USA
[6] Kyushu University, Fukuoka, Japan
[7] Institute for Climate and Atmospheric Science, ETH Zurich, Zurich, Switzerland
[8] CICERO Center for International Climate and Environmental Research, Oslo, Norway
[9] Finnish Meteorological Institute, Atmospheric Research Centre of Eastern Finland, Kuopio, Finland

*Correspondence to*: George Jordan (george.jordan@metoffice.gov.uk)

**Abstract.** For over 6-months, the 2014–2015 effusive eruption at Holuhraun, Iceland injected considerable amounts of sulphur dioxide ($SO_2$) into the lower troposphere with a daily rate of up to one-third of the global emission rate causing extensive air pollution across Europe. The large injection of $SO_2$, which oxidises to form sulphate aerosol ($SO_4^{2-}$), provides a natural experiment offering an ideal opportunity to scrutinise state-of-the-art general circulation models' (GCMs) representation of aerosol-cloud interactions (ACIs). Here we present Part 1 of a two-part model inter-comparison using the Holuhraun eruption as a framework to analyse ACIs. We use $SO_2$ retrievals from the Infrared Atmospheric Sounding Interferometer (IASI) instrument and ground-based measurements of $SO_2$ and $SO_4^{2-}$ mass concentrations across Europe, in conjunction with a trajectory analysis using the Hybrid Single Particle Lagrangian Integrated Trajectory (HYSPLIT) model, to assess the spatial and chemical evolution of the volcanic plume as simulated by 5 GCMs and a chemical transport model (CTM). IASI retrievals of plume altitude and $SO_2$ column load reveal that the volcanic perturbation is largely contained within the lower troposphere. Compared to the satellite observations, the models capture the spatial evolution and vertical variability of the plume reasonably well, although the models often overestimate the plume altitude. HYSPLIT trajectories are used to attribute to Holuhraun emissions 111 instances of elevated sulphurous surface mass concentrations recorded at European Monitoring and Evaluation Programme (EMEP) stations during September and October 2014. Comparisons with the simulated concentrations show that the modelled ratio of $SO_2$-to-$SO_4^{2-}$ during these pollution episodes is often underestimated and overestimated for the young and mature plume respectively. Models with finer vertical resolutions near the surface are found to better capture these elevated

sulphurous ground-level concentrations. Using an exponential function to describe the decay of observed surface mass concentration ratios of $SO_2$-to-$SO_4^{2-}$ with plume age, the in-plume oxidation rate constant is estimated as $0.032 \pm 0.002$ h$^{-1}$ ($1.30 \pm 0.08$ day $e$-folding time) with a near-vent ratio of $25 \pm 5$ [µgm$^{-3}$ of $SO_2$ / µgm$^{-3}$ of $SO_4^{2-}$]. The majority of the corresponding derived modelled oxidation rate constants are lower than the observed estimate. This suggests that the representation of the oxidation pathway/s in the simulated plumes is too slow. Overall, despite their coarse spatial resolutions, the 6 models show reasonable skill in capturing the spatial and chemical evolution of the Holuhraun plume. This capable representation of the underlying aerosol perturbation is essential to enable the investigation of the eruption's impact on ACIs in the second part of this study.

## 1 Introduction

The Bárðarbunga volcanic system in Iceland began experiencing noteworthy sequences of earthquakes on 16[th] August 2014 (Ágústsdóttir et al., 2016; Gudmundsson et al., 2014; Sigmundsson et al., 2015). This seismic activity created an effusive eruption at Holuhraun (64.85°N, 16.83°W) from 31[st] August 2014 to 27[th] February 2015. The resulting flow of lava is estimated to have covered 84 km$^2$ with an average discharge rate of 90 m$^3$s$^{-1}$ making it the largest effusive eruption in Iceland since the 1783-1784 Laki eruption (Pedersen et al., 2017). Ground-based observation estimates suggest the Holuhraun eruption released a total of 9.6–11.8 Mt of sulphur dioxide ($SO_2$) with little tephra (Gíslason et al., 2015; Pfeffer et al., 2018). These emissions represent up to 215 times the amount of Icelandic anthropogenic $SO_2$ emissions for 2019 (Keller et al., 2022) and approximately one tenth of the current global annual anthropogenic $SO_2$ emissions (e.g. Aas et al., 2019). During these months of intense degassing, several studies using ground-based observations and remote sensing suggest that the volcanic plume reached altitudes of 1–4 km (Arason et al., 2015; Carboni et al., 2019a; Flower and Kahn, 2020; Pfeffer et al., 2018). This release of $SO_2$ adversely affected air quality over large parts of Iceland often exceeding World Health Organization (WHO) surface concentration limits (Gíslason et al., 2015; Ilyinskaya et al., 2017; Schmidt et al., 2015). Such high rates of $SO_2$ emission into a relatively pristine, unpolluted environment provide an ideal opportunity to observe aerosol-cloud interactions (ACIs) and elucidate aerosol impacts on the climate system (e.g. Breen et al., 2021; Chen et al., 2022; Gettelman et al., 2015; Haghighatnasab et al., 2022; Malavelle et al., 2017, McCoy and Hartmann, 2015, Toll et al., 2017).

Remote sensing data estimates of $SO_2$ concentrations, a common tool to monitor the evolution of volcanic plumes, show that the September and October meteorological conditions transported the plume across Europe including the Fennoscandian Peninsula (Grahn et al., 2015; Ialongo et al., 2015), Belgium, northern France (Boichu et al., 2016), the UK, Ireland, the Netherlands (Schmidt et al., 2015) and Germany (Steensen et al., 2016). Previous studies have combined satellite data with ground-based observations and trajectory modelling to attribute local pollution events to the Holuhraun eruption and investigate the plume characteristics (e.g. Boichu et al., 2019; Schmidt et al., 2015; Twigg et al., 2016). Understanding such characteristics, particularly those that are hazardous to health (i.e. fine particulate matter), is important for air quality

monitoring and attribution of any exceedances of statutory thresholds (Heaviside et al. 2021; Stewart et al., 2022). However, most studies assessing Holuhraun impacts on air quality focus only on pollution experienced at a few ground-based stations across small geographical areas and the need to increase the quantity of air monitoring stations investigated has been noted

previously (Schmidt et al., 2015; Twigg et al., 2016).

In addition to observational evidence, many studies have explored the impacts of the Holuhraun eruption using numerical models to simulate the transport of the volcanic $SO_2$ emissions. For instance, the Icelandic Meteorological Office employed the CALPUFF dispersion model for near-time probabilistic hazard mapping (Barsotti, 2020) and to support aviation safety

decisions (Barsotti et al., 2020) following the eruption. Boichu et al. (2016) and Steensen et al. (2016) used the EMEP MSC-W and the CHIMERE chemical transport models (CTMs) respectively to explore the far-range air pollution effects caused by the eruption, whereas Schmidt et al. (2015) used the NAME dispersion model to do so. Recently, Haghighatnasab et al. (2022) analysed the results from high resolution simulations performed with the ICON model in its numerical weather prediction mode to explore the impact of the aerosol perturbation introduced by Holuhraun on cloud properties. This impact was also

examined in earlier works using general circulation models (GCMs) of coarser resolutions (CAM5 – Gettelman et al., 2015; HadGEM3, CAM5, and a NorESM variant – Malavelle et al., 2017). Considering the opportunity the Holuhraun eruption presents to assess the modelling capability of GCMs in capturing aerosol-cloud interactions, the number of GCM studies of the Holuhraun eruption to date is surprisingly low.

Here we present results from an inter-model comparison two-part study of the volcanic plume and its interactions with clouds within the vicinity of the Holuhraun eruption (44°N to 80°N, 60°W to 30°E) during September and October 2014. Participation in the study was organised through the AeroCom initiative (Schulz et al., 2006). We begin with a comparison of the volcanic $SO_2$ plume evolution between remote sensing observations and simulations of the eruption from 5 GCMs and a CTM. The analysis specifically focuses on the plume spatial distribution, plume altitude, and the total $SO_2$ mass burden. We further

investigate the numerical models' capability to simulate the Holuhraun eruption by comparing modelled $SO_2$ and sulphate ($SO_4^{2-}$) concentrations with in situ surface observations from 25 long-term monitoring stations from the European Monitoring and Evaluation Programme (EMEP) network. By using remote sensing and ground-based observations, in conjunction with trajectory modelling, we attribute sulphurous pollution events to Holuhraun emissions and assess the skill of the numerical models in capturing these episodes. Finally, this refined catalogue of volcanically influenced pollution events is used to

estimate the rate at which $SO_2$ oxidises to $SO_4^{2-}$ within both observed and modelled volcanic plumes. We conclude with a discussion of whether the models simulate the observed Holuhraun aerosol perturbation with sufficient fidelity; a prerequisite for using a model to understand the climatic impacts caused by the eruption via ACIs (see Part 2 of this study).

## 2 Methodology

We provide a brief description of the remote sensing and in situ observations that are used to assess the numerical models, the numerical models themselves, and the HYSPLIT trajectory modelling framework used to evaluate the pollutant transport of the observed local sulphurous events.

### 2.1 Satellite Observations

Retrievals of volcanic $SO_2$ from satellite instrumentation typically use either measurements in the ultra-violet (e.g. Ozone
Mapping and Profiler Suite – Nadir Mapper (OMPS-NM); Carn et al., 2015; Li et al., 2017; Wells et al., 2023; TROPOspheric Monitoring Instrument (TROPOMI); de Leeuw et al., 2021; Theys et al., 2017; Global Ozone Monitoring Experiment-2 (GOME-2); Twigg et al., 2016) or infra-red (e.g. Infrared Atmospheric Sounding Interferometer (IASI); Clarisse et al., 2008, 2010, Haywood et al., 2010, de Leeuw et al., 2021) region of the electromagnetic spectrum. Here we use IASI measurements as they have proved valuable in monitoring the evolution of volcanic plumes in both the stratosphere (e.g. Haywood et al.,
2010; de Leeuw et al., 2021) and the troposphere (e.g. Athanassiadou et al., 2016; Malavelle et al., 2017). Specifically, we use data from IASI retrievals on the MetOp-A and MetOp-B satellites produced by the University of Oxford as part of the NERC Centre for the Observation and Modelling of Earthquakes, Volcanoes and Tectonics (COMET) (Carboni et al., 2019a, b).

$SO_2$ column load and plume height are derived by applying the IASI retrieval algorithm of Carboni et al. (2012, 2016) to level
1C data from the EUMETSAT and CEDA archive. The IASI $SO_2$ retrieval is performed only on pixels where the underlying $SO_2$ detection scheme returns a positive result. The detection scheme is a linear retrieval where a positive result is defined as when the free parameter, the $SO_2$ column load, exceeds a defined threshold. This threshold is set substantially greater than the standard deviation of the assumed Gaussian distribution describing the background atmospheric concentration of $SO_2$. Consequently, a positive result is exceedingly likely to be significantly different to the background, and not a consequence of
instrumental noise or climatological variations (see details in Walker et al. 2011, 2012). The threshold defined for the Holuhraun eruption in Carboni et al. (2019a) is 0.49 effective DU.

An iterative optimal estimation retrieval using forward modelled spectra is applied to pixels with a positive detection result. This retrieval uses all channels within 1000–1200 cm$^{-1}$ and 1300–1410 cm$^{-1}$ (the 7.3 µm and 8.7 µm $SO_2$ bands respectively)
and assumes a Gaussian vertical $SO_2$ profile to return the $SO_2$ column load (DU) and height (mb) which the retrieval algorithm subsequently converts to km using European Centre for Medium-Range Weather Forecasts (ECMWF) meteorological profiles. The algorithm provides a comprehensive pixel-by-pixel error estimate on the retrieved parameters that is derived from an error covariance matrix computed using the differences between the measured IASI spectra and the simulated spectra (driven by ECMWF data). This means uncertainty due to imperfect knowledge of non-$SO_2$ atmospheric conditions (e.g. cloudiness,
vertical distribution of constituents) and imperfect radiative transfer simulations are addressed (see details in Carboni et al.,

2012). The thermal contrast between the plume and surface heavily influences the retrieval error such that retrievals of $SO_2$ plumes centred at lower altitudes have higher uncertainties. Note that the IASI retrieval algorithm can miss parts of the $SO_2$ plume, such as when overlaying clouds are present or under conditions of negative thermal contrast, and so the IASI $SO_2$ column load and mass burdens presented here should be considered an approximate minimum.


This study maps data from individual IASI overpasses to a regular 1.0° x 1.0° latitude-longitude grid using a nearest neighbours with Gaussian weighting approach. The decision to weight closer neighbouring pixels allows retention of plume characteristics which can change abruptly over small spatial scales. The individual gridded overpass data are grouped into bidaily intervals (AM: 03:30–15:30 UTC, PM: 15:30–03:30 UTC) with overlapping cells averaged. Linear interpolation is used to estimate

missing values in the gridded output that result from orbital gaps and/or pixels failing quality control. Each bidaily regridded IASI $SO_2$ column load and altitude maps are visually inspected to ensure no obvious artefacts exist within the Holuhraun vicinity.

## 2.2 Surface Observations

Since the early 1970s, the EMEP network has monitored air pollution and surface deposition across Europe at ground-level

stations outside of notable conurbations where significant sources of local pollution are minimised, thus creating a comprehensive database useful for assessing long-range transportation of a plethora of pollutants (Tørseth et al., 2012). The use of EMEP stations to evaluate model output has proven fruitful previously (e.g. Hardacre et al., 2021; Mulcahy et al., 2020). This study only considers EMEP stations that provide both $SO_2$ and $SO_4^{2-}$ surface mass concentration measurements at the same temporal sampling frequency during September and October 2014. This criterion results in 25 stations located across 12

countries being selected for this study (see Table 1). The observations include hourly and daily measurements made using online ion chromatography and filter-pack measurements respectively, with the former to a precision of 0.001 $\mu gm^{-3}$ and the latter to either 0.01 $\mu gm^{-3}$ or 0.001 $\mu gm^{-3}$ depending on the station. The hourly and daily sampling midpoints are centred on 30 minutes past the hour and on the hour respectively. Further details on the instruments and sampling techniques are provided in the EMEP Standard Operating Protocol (NILU, 2014). This study screens out invalid and missing measurements in

accordance with the EMEP data quality flags (NILU, 2020). For each station monthly surface mass concentration climatologies for $SO_2$ and $SO_4^{2-}$ are calculated from the temporal coverage listed in Table 1. For a given station this coverage may differ for the two chemical species. Subsequently, the combined total sulphur content climatologies are only calculated across periods where the temporal coverages align (e.g. 1988–2017 for Aspvreten and 2006–2020 for Irafoss). Here we define a significant sulphurous pollution event as when the surface mass concentration of the total sulphur content observed exceeds the 90th

percentile of the corresponding monthly climatological value. Note that the number of EMEP stations carrying out $SO_2$ and $SO_4^{2-}$ measurements has fallen since the late 2000s due to the reduced need to monitor the declining sulphur emissions from anthropogenic sources (Boichu et al., 2019; Schmidt et al., 2015).

**Table 1**: Details of the 25 EMEP stations explored in this study. These stations are shown geographically in Fig. 1.

| Station name (EMEP code) | Country | Sampling details | | | Coordinates | | Trajectory details | |
| --- | --- | --- | --- | --- | --- | --- | --- | --- |
| | | Instrument type/s | Frequency | Temporal coverage | Lat. lon. (°N, °E) | Alt. (m AMSL) | Starting height (m AGL) | Bounding radius (km) |
| Anholt (DK0008R) | Denmark | Filter-3pack | Daily | 1989-2020 | (56.72°, 11.52°) | 40 | 100 | 380 |
| Aspvreten (SE0012R) | Sweden | Filter-3pack Filter-2pack Filter-1pack | Daily | 1984-2017 (SO$_2$ from 1988) | (58.80°, 17.38°) | 20 | 100 | 440 |
| Auchencorth Moss (GB0048R) | Scotland | Online Ion Chroma. | Hourly | 2007-2020 | (55.79°, -3.24°) | 260 | 250 | 320 |
| Birkenes II (NO0002R) | Norway | Filter-3pack | Daily | 2010-2020 | (58.39°, 8.25°) | 219 | 100 | 320 |
| Bredkälen (SE0005R) | Sweden | Filter-3pack Filter-2pack Filter-1pack | Daily | 1980-2020 (SO$_2$ from 1992) | (63.85°, 15.33°) | 404 | 100 | 380 |
| Harwell (GB0036R) | England | Online Ion Chroma. | Hourly | 2009-2015 (SO$_2$ from 2011) | (51.57°, -1.32°) | 137 | 100 | 380 |
| Hurdal (NO0056R) | Norway | Filter-3pack | Daily | 1997-2020 | (60.37°, 11.08°) | 300 | 100 | 320 |
| Irafoss (IS0002R) | Iceland | Filter-2pack Filter-1pack | Daily | 1980-2020 (SO$_2$ from 2006) | (64.08°, -21.02°) | 66 | 100 | 72 |
| Kårvatn (NO0039R) | Norway | Filter-3pack Filter-2pack | Daily | 1980-2020 | (62.78°, 8.88°) | 210 | 100 | 320 |
| Leba (PL0004R) | Poland | Filter-2pack Filter-1pack | Daily | 1993-2020 | (54.75°, 17.53°) | 2 | 100 | 500 |
| Neuglobsow (DE0007R) | Germany | Filter-3pack Filter-1pack | Daily | 1981-2018 (SO$_2$ from 2000) | (53.17°, 13.03°) | 62 | 100 | 500 |
| Pallas Matorova (FI0036R) | Finland | Filter-3pack Filter-2pack | Daily | 1996-2020 | (68.00°, 24.24°) | 340 | 250 | 440 |
| Preila (LT0015R) | Lithuania | Filter-3pack Filter-2pack | Daily | 1991-2020 (SO$_2$ from 1996) | (55.38°, 21.03°) | 5 | 250 | 500 |
| Råö (SE0014R) | Sweden | Filter-3pack | Daily | 2002-2020 | (57.39°, 11.91°) | 5 | 100 | 380 |
| Risoe | Denmark | Filter-3pack | Daily | 2011-2020 | (55.69°, 12.09°) | 3 | 100 | 440 |

| | | | | | | | | |
|---|---|---|---|---|---|---|---|---|
| (DK0012R) | | | | | | | | |
| Rucava (LV0010R) | Latvia | Filter-2pack Filter-1pack | Daily | 1986-2020 ($SO_2$ from 1990) | (56.16°, 21.17°) | 18 | 100 | 500 |
| Schauinsland (DE0003R) | Germany | Filter-3pack | Daily | 2000-2018 | (47.91°, 7.91°) | 1205 | 550 | 500 |
| Tange (DK0003R) | Denmark | Filter-3pack Filter-2pack | Daily | 1978-2020 | (56.35°, 9.60°) | 13 | 100 | 380 |
| Tustervatn (NO0015R) | Norway | Filter-3pack Filter-2pack | Daily | 1980-2020 | (65.83°, 13.92°) | 439 | 100 | 320 |
| Utö (FI0009R) | Finland | Filter-3pack Filter-2pack Filter-1pack | Daily | 1980-2020 ($SO_2$ from 1991) | (59.78°, 21.38°) | 7 | 100 | 440 |
| Valentia Observatory (IE0001R) | Ireland | Filter-3pack Filter-2pack | Daily | 1980-2020 | (51.94°, -10.24°) | 11 | 100 | 320 |
| Vavihill (SE0011R) | Sweden | Filter-3pack Filter-2pack Filter-1pack | Daily | 1984-2015 ($SO_2$ from 1992) | (56.02°, 13.15°) | 175 | 150 | 440 |
| Virolahti II (FI0017R) | Finland | Filter-3pack Filter-2pack | Daily | 1989-2014 ($SO_2$ from 1991) | (60.53°, 27.69°) | 4 | 100 | 500 |
| Waldhof (DE0002R) | Germany | Filter-3pack | Daily | 2000-2018 ($SO_4^{2-}$ from 2005) | (52.80°, 10.76°) | 74 | 100 | 440 |
| Zeppelin Mountain (NO0042G) | Norway | Filter-3pack | Daily | 1990-2020 | (78.91°, 11.89°) | 474 | 350 | 440 |


## 2.3 Numerical Model Simulations

Included in this study are Holuhraun eruption simulations by 5 GCMs: UKESM1.0, HadGEM3-GA7.0, MIROC6.1-
SPRINTARS, ECHAM6.3-HAM2.3, and ECHAM6.3-HAM2.3-P3. Simulations are performed using the atmosphere-only component at a global scale (AMIP-style). To help clearly discriminate between signal and noise, the modelled horizontal winds and potential temperature are constrained ("nudged") to ERA-Interim reanalysis data (Dee et al., 2011) on a 6-hourly time scale, and use monthly observational datasets to prescribe sea surface temperature and sea ice boundary conditions (e.g. HadISST, Rayner et al., 2003). All other modelled variables evolve physically and dynamically as their setup dictates and are
subject to the parameterisations in play. Also included in our inter-model comparison is OsloCTM3, a global CTM. Unlike GCMs, CTMs do not simulate atmospheric dynamics explicitly, instead OsloCTM3 uses pre-calculated 3-hourly meteorological fields from ECMWF forecasts produced daily with a 12-hourly spin-up starting from ERA-Interim reanalysis. All numerical model simulations assume the eruption starts on 31$^{st}$ August 2014 and that the Holuhraun $SO_2$ emissions are distributed equally in the vertical for grid cells between 0.8 km and 3 km in the column containing the eruption vent following
the magnitude and altitude profile of emissions described in Malavelle et al. (2017). All models include additional background $SO_2$ emissions from anthropogenic and natural sources. The simulations are continued from multiyear control simulations. All model output is regridded to a common regular 1.0° x 1.0° latitude-longitude grid using linear interpolation. Details specific to individual numerical models and key references can be found in Table 2.

**Table 2:** Details of the numerical models used in this study.

| Model name | Modelling centre | Chemistry/ Aerosol module | Resolution | | | Constraining / nudging data | References |
|---|---|---|---|---|---|---|---|
| | | | Atmospheric grid (lat. x lon.) | Surface layer thickness (m) | Levels within 3 km (AMSL) | | |
| UKESM1.0 | Met Office Hadley Centre, UK | UKCA-Mode | N96 L85 (1.25° x 1.875°) | 20 | 20 | ERA-Interim | Mulcahy et al., 2020 Sellar et al., 2019 |
| HadGEM3-GA7.0 | Met Office Hadley Centre, UK | UKCA-Mode | N96 L85 (1.25° x 1.875°) | 20 | 20 | ERA-Interim | Mulcahy et al., 2020 Walters et al., 2019 |
| MIROC6.1-SPRINTARS | Research Institute for Applied Mechanics, Kyushu University, Japan | SPRINTARS | T213 L40 (0.5625° x 0.5625°) | 45 | 13 | ERA-Interim | Tatebe et al., 2019 Takemura et al., 2000, 2005, 2009 |
| ECHAM6.3-HAM2.3 | University of Oxford, UK | HAM (Default cloud microphysics scheme) | T63 L47 (1.875° x 1.875°) | 68 | 9 | ERA-Interim | Neubauer et al., 2019 Stevens et al., 2013 Tegen et al., 2019 |
| ECHAM6.3-HAM2.3-P3 | ETH Zurich, Zurich, Switzerland | HAM-P3 (P3 cloud microphysics scheme) | T63 L47 (1.875° x 1.875°) | 68 | 9 | ERA-Interim | Dietlicher et al., 2018 Neubauer et al., 2019 Stevens et al., 2013 Tegen et al., 2019 |
| OsloCTM3 | CICERO Center for International Climate Research, Norway | Stratospheric and tropospheric chemistry schemes | N80 L60 (2.25° x 2.25°) | 10 | 16 | ECMWF forecasts (initiated with ERA-Interim) | Berntsen et al., 1997 Lund et al., 2018 Søvde et al., 2012 |

## 2.4 Backward Trajectories

Lagrangian modelling has been used previously to study the long-range transport of Holuhraun pollutants (e.g. Boichu et al., 2019; Schmidt et al., 2015). Here the origin, age, and travel distance of air parcels associated with sulphurous pollution events detected in the EMEP network observations are estimated using backward single-particle trajectories generated by the Hybrid Single-Particle Lagrangian Integrated Trajectory (HYSPLIT) model developed by the National Ocean and Atmospheric Administration (NOAA) Air Resources Laboratory (Stein et al., 2015). We use 6-hourly ERA-Interim reanalysis data interpolated to an hourly resolution and regridded onto a 1.0° x 1.0° latitude-longitude grid as the meteorological input to the HYSPLIT model; a choice made to keep the driving meteorological data consistent between the trajectory analysis, and the nudging of the GCMs and CTM. Beginning on 1st September 2014 00:00 UTC, every hour at each EMEP station a new 27-member ensemble of 10-day backward trajectories is initiated at the coordinates and starting heights listed in Table 1 until the 31st October 2014 (a total of 1464 ensembles for each station) with locations along each trajectory saved hourly. The ERA-Interim reanalysis data for each ensemble is offset by a fixed grid factor, a maximum of 1.0° of latitude/longitude in the horizontal and 0.01 sigma units in the vertical, and so all the possible meteorological offsets result in the 27 members within each ensemble. As our ensembles are initiated at the beginning of every hour, pollution events observed at a daily resolution with a sampling midpoint centred on the hour have 25 ensembles available for analysis (12 hours either side of the midpoint and the midpoint itself), whilst those events observed hourly with a sampling midpoint centred 30 minutes past the hour will have two ensembles (the bounding hours of the midpoint). This equates to a total of 675 and 54 individual trajectories respectively to evaluate the pollutant transport of each event.

A limiting factor with using a backward trajectory analysis is that the trajectories are not expected to arrive exactly at the eruption vent. Subsequently, a domain must be defined that sets the bounds as to whether a trajectory is deemed close enough to be attributed to the volcano. Defining this domain can be done visually through satellite imagery (e.g. Pardini et al., 2017) or by a statistical analysis (e.g. Hughes at al., 2012). Here we adopt the latter, defining multiple 3-D bounding cylinders centred on 64.85°N, 16.83°W with a height above mean sea level of 4.5 km and various bounding radii (see dotted regions in Fig. 1). The cylinder height is based on the maximum plume altitudes within the literature (Arason et al., 2015; Carboni et al., 2019a; Flower and Kahn, 2020; Pfeffer et al., 2018), whilst bounding radii are dependent on the distance from the eruption that the trajectories are initiated at. Trajectories released from stations distanced 1200–1500 km, 1500–1800 km, 1800–2100 km, and 2100–2400 km from Holuhraun are subject to radii of 320 km, 380 km, 440 km, and 500 km respectively. These values are based on the positional error of a trajectory being approximately 10–30% of the total distance travelled (Stohl, 1998). A special case is made for the Irafoss station due to its close proximity to Holuhraun (~200 km). Over this distance, due to the finite hourly resolution of the trajectories and magnitude of local wind speeds, a trajectory is likely to travel further in a single time step than the estimate obtained from the "10–30% distance travelled" method. Hence, even an ideal trajectory passing directly

over the eruption may not be outputted within the bounding radius. Consequently, to ensure a near 100% likelihood these ideal trajectories are captured, we define the Irafoss radius as the 99[th] percentile of the September and October ERA-Interim reanalysis horizontal wind speeds of the grid cells containing the horizontal eruption location and with midpoints below the cylinder height (see Supplement, S1).

This study attributes an observed sulphurous pollution event to volcanic emissions from Holuhraun when at least 25% of the released trajectories pass through the relevant 3-D bounding cylinder, equating to a minimum of 167 and 14 trajectories for daily and hourly sampled events respectively. Whilst this threshold could be considered low, allowing for other sources to contribute to the sulphurous pollution detected, the sheer volume of Holuhraun emissions within the region versus other sources during September and October 2014, and the rural location of the surface stations, gives confidence that this threshold is

sufficient. For a given pollution event, we average the transport time and travel distance of the individual trajectories attributed to Holuhraun at their point of closest approach to the eruption to estimate the age and distance travelled by the plume. The error in the plume age is estimated as the larger value of either the standard error of the trajectories sampled or the trajectory temporal resolution (1 h).

Like all frameworks based on single-particles trajectories, our trajectory analysis is subject to the inherent uncertainty associated with individual trajectories (Stohl, 1998) with the uncertainty in the input meteorology often regarded as the dominant contribution (Engström and Magnusso, 2009; Gebhart et al., 2005; Harris et al., 2005). Here we minimise this main cause of uncertainty by perturbing the meteorology for each ensemble member. However, other uncertainties, such as the choice of meteorological dataset and/or trajectory model, and the exclusion of turbulence, have not been accounted for in our

trajectory framework. Although these uncertainties are relevant, the focus of this study is to inter-compare numerical models consistently rather than through a rigorous dispersion exercise, and so they will not be considered further. Despite the simplicity of our single-particle trajectory framework, similar methods have been applied successfully to surface monitoring stations (e.g. Nieminen et al., 2015; Räty et al., 2023; Väänänen et al., 2013).


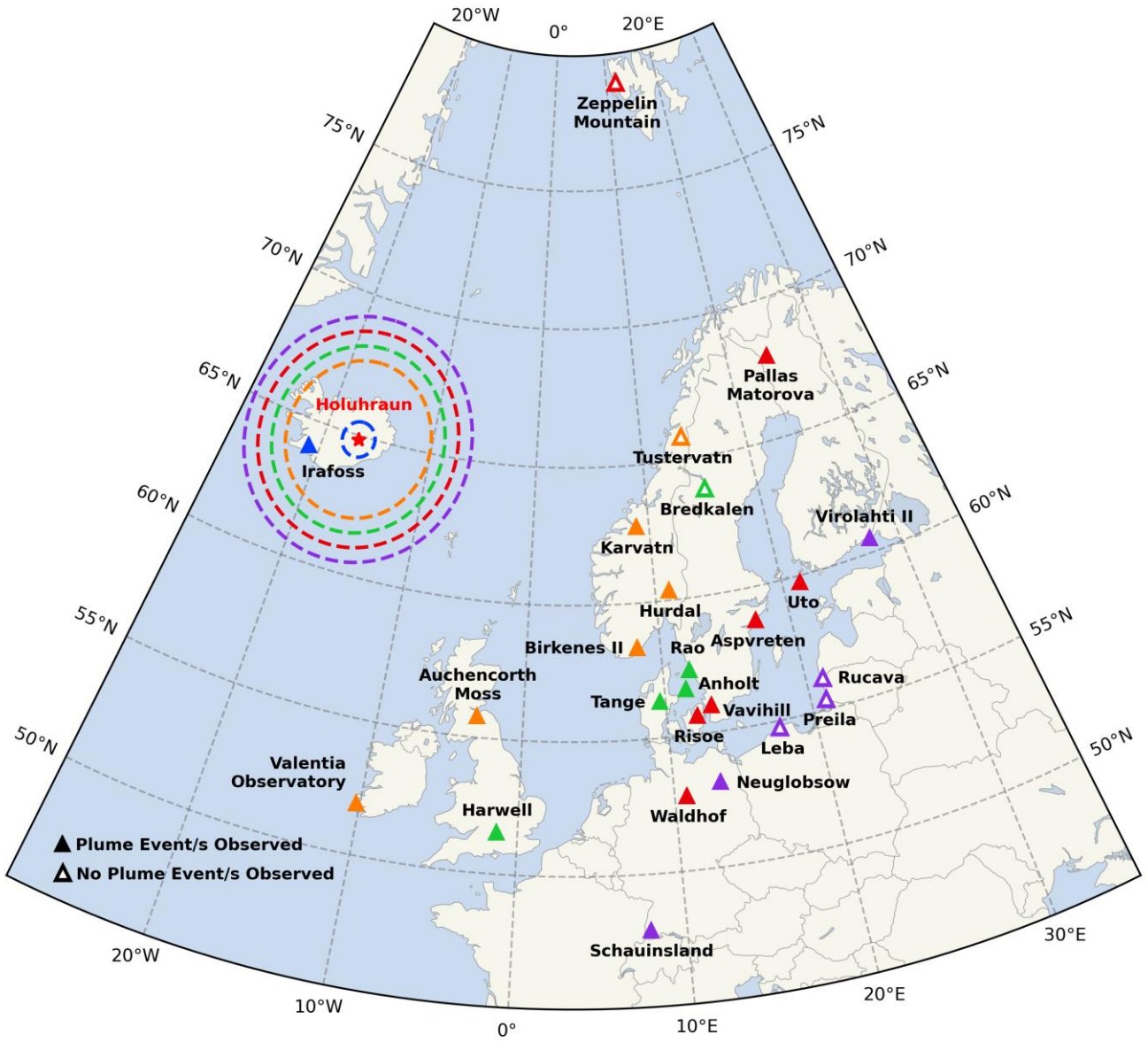

**Figure 1:** Map of the 25 EMEP stations explored in this study. Stations marked with filled triangles experienced at least one sulphurous pollution event between 1ˢᵗ September to 31ˢᵗ October 2014 attributed to Holuhraun emissions, whereas stations marked with unfilled triangles did not. A red star indicates the location of the Holuhraun eruption (64.85°N, 16.83°W) with the surrounding dashed lines outlining the horizontal boundaries of the Holuhraun bounding areas defined in this study. From the inner circle outwards, the radii are: 72 km, 320 km, 380 km, 440 km, and 500 km. Colouring links a station to the bounding area it is subject to.

## 3. SO₂ Plume Spatial Distribution

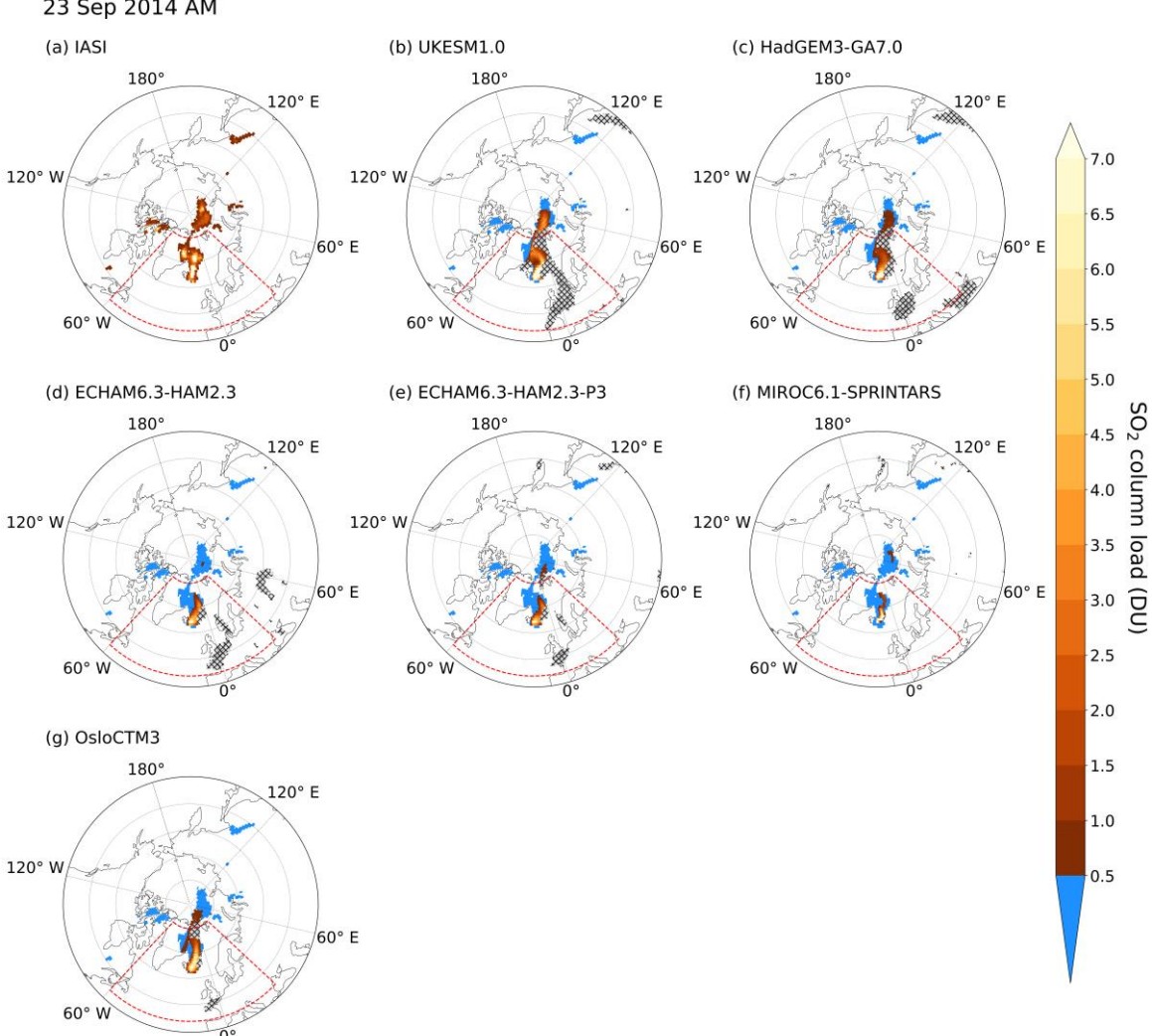

**Figure 2:** SO₂ column load from (a) Infrared Atmospheric Sounding Interferometer (IASI) retrievals and (b, c, d, e, f, g) model simulations of the Holuhraun eruption on the 23ʳᵈ September 2014 AM. IASI data below 0.5 Dobson units (DU) are masked to identify the observed plume extent. Model output is sampled at 09:30 UTC. Non-hatched areas show model output within the observed plume extent with the distinct blue highlighting model values below the 0.5 DU threshold. Hatched areas show model output outside the observed plume extent that are above the 0.5 DU threshold. Both IASI and model data have been regridded onto the same 1.0° x 1.0° latitude-longitude grid. Red dashed areas represent the Holuhraun region defined in this study. An animation of September and October 2014 can be found in the Supplement (S2).

The SO$_2$ column load from IASI retrievals and model simulations on the 23$^{rd}$ September 2014 AM are displayed in Fig. 2 and an animation spanning September and October 2014 can be found in the Supplement (S2). We mask IASI SO$_2$ column load retrievals below 0.5 DU to identify the observed horizontal extent of the plume. This threshold sufficiently exceeds the region's typical background SO$_2$ column load of approximately 0.1 DU; a background derived using the 2007–2009 September mean SO$_2$ mass burden of a similar geographical region given in Schmidt et al. (2015). Applying such a threshold ensures enough SO$_2$ from other sources are screened out whilst not removing data associated with the main volcanic plume. To enable a comparison with the IASI retrievals, the model output is regridded onto the same 1.0° x 1.0° latitude-longitude grid and is sampled within the observed plume extent at 09:30 UTC and 21:30 UTC for the AM and PM grouped retrievals respectively. Areas within the observed plume extent that the models fail to capture (i.e. values below 0.5 DU) are shown in a distinct blue.

Within the Holuhraun region (red dashed area) in Fig. 2, UKESM1.0, HadGEM3-GA7.0 and OsloCTM3 perform well in capturing the observed plume extent with minimal blue areas present, whilst the ECHAM variants and MIROC6.1-SPRINTARS roughly capture 50% and 30% respectively. These performances largely hold true across September and October as evident in the animation. Although, due to the binary nature of the observed plume extent masking, no magnitude on how far the modelled values lie below 0.5 DU is given and so this metric should not be considered a sole indicator of model performance. In addition, we sample the models outside the observed plume extent when the modelled SO$_2$ column load exceeds 0.5 DU (hatched areas). In Fig. 2 all models except MIROC6.1-SPRINTARS simulate the plume outside the observed area over parts of Western Europe. These areas potentially arise due to IASI retrieval limitations causing parts of the plume to be missed (e.g. cloud cover, high latitude, swath width) or due to high levels of background SO$_2$ emissions in the models (e.g. volcanic activity from Mt. Etna, anthropogenic activity). In September, for UKESM1.0, HadGEM3-GA7.0, and OsloCTM3 the hatched areas tend to dominate the blue areas suggesting that their modelled plume areas are greater than the IASI retrievals, whereas the opposite is true for the remaining models. In October, all models largely show a greater modelled plume area than observed, yet this is partly due to the low IASI coverage across this period.

Overall, Fig. 2 and the animation make it apparent that the Holuhraun eruption is observed and modelled as the main source of SO$_2$ in our region of interest. Contributions from high background SO$_2$ sources are minimal relative to the total regional SO$_2$ and occur either outside or just within the outer bounds of the region, so are unlikely to substantially influence this study. Both visualisations show that the models capture the general features of the observed plume, particularly the dispersion over the Fennoscandian Peninsula and the UK during September, suggesting that nudging the models to ERA-Interim reanalyses gives credence to the models' ability to accurately simulate the plume dispersion despite their coarse resolution. It is worth noting that the animation shows possible regridding artefacts in the IASI retrievals on the 11$^{th}$, 12$^{th}$, and 17$^{th}$ September PM. These artefacts occur outside our region of interest and so will not be considered further here.

## 4. SO₂ Plume Height and Mass Burden

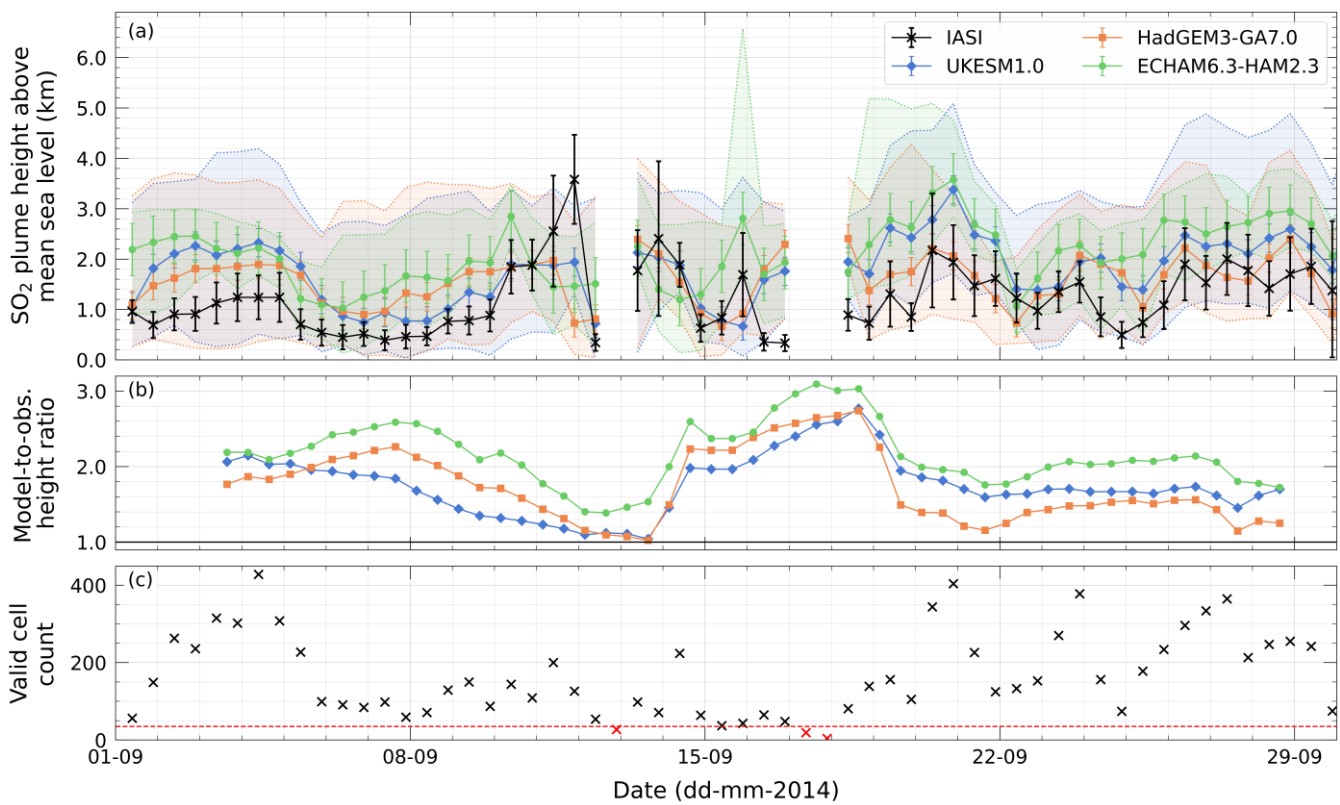

**Figure 3:** (a) Bidaily temporal evolution and (b) 5-day rolling mean of IASI retrieved and modelled SO₂ plume heights across the Holuhraun region (44°N to 80°N, 60°W to 30°E) for September 2014. Black crosses and error bars represent the regional mean IASI SO₂ plume height and associated error. Models are sampled only within the observed plume extent. The coloured lines and error bars represent the regional mean central height of the modelled SO₂ plume and associated error, whilst an envelope indicates the regional mean height of the top and bottom of the modelled SO₂ plume (see text). The 5-day rolling means tolerate a maximum of two missing data points. (c) Number of grid cells used to calculate the regional means. The red dashed line indicates the minimum number of grid cells deemed sufficient to enable a fair comparison between observed and modelled. Both IASI and model data have been regridded onto the same 1.0° x 1.0° latitude-longitude grid.

The average observed and modelled SO₂ plume heights across the Holuhraun region for September and October 2014 are shown in Fig. 3a and the Supplement (S3) respectively, with monthly values provided in Table 3. The IASI retrievals show that the observed SO₂ plume height, specifically the central height of a Gaussian SO₂ vertical profile, exists primarily (~75%) between 0.8–2.5 km above mean sea level, very rarely exceeding 3 km, showing that the volcanic perturbation to the region is contained well within the lower troposphere. The modelled SO₂ plume vertical extents are determined from model output that

has been subject to the same sampling used in Sect. 3, plus an additional masking in the vertical of grid cells with $SO_2$ mass concentrations below 4.5 $\mu gm^{-3}$; a threshold based on the clean air $SO_2$ concentration of 1 ppbv (roughly 3 $\mu gm^{-3}$ at 2–3 km) given in Theys et al. (2013). From the remaining grid cells with sufficient concentrations in the observed plume extent, the modelled central $SO_2$ plume height, represented by a solid line, is calculated as the regional mean of the heights of the grid cells containing the maximum $SO_2$ concentration in each column. The associated error is based on the typical vertical resolution of the model between 2–4 km above mean sea level. The top and bottom of the modelled $SO_2$ plume, illustrated by the outer dotted lines enclosing the envelope, are the regional averages of the maximum and minimum heights of the sufficiently polluted grid cells. Fig. 3c shows the underlying number of data points contributing to the regional means. Regional means calculated from a number of data points below the minimum threshold of 35 (red dashed line) are deemed inadequate for a fair comparison and do not contribute to the rolling means shown in Fig. 3b. Note, from the output provided to this experiment, ECHAM6.3-HAM2.3-P3 and MIROC6.1-SPRINTARS $SO_2$ plume heights can only be compared with IASI retrievals at a monthly resolution, whereas no comparison is possible for OsloCTM3.

Generally, UKESM1.0, HadGEM3-GA7.0, and ECHAM6.3-HAM2.3 overestimate the central plume height across September as clearly evidenced in the 5-day rolling mean model-to-observed ratio. This overestimation is greatest during the third week, particularly on the 16[th] and 18[th] September. This feature may be a consequence of the vertical winds in the models not being constrained and/or additional variability in the momentum flux during the eruption which is not accounted for in the prescribed emission profile used in the models. The observed variability in the plume height is largely well represented in the three models with all peaks, aside from the 11[th] September PM, captured within error whilst the observed central height is very rarely found outside the modelled vertical extents. The bottoms of the modelled vertical extents are close to the surface suggesting that ground-based stations within the region are likely to experience moments of sulphurous pollution due to the eruption. The performance of the three models during October is similar (see Supplement, S3), although the limited IASI retrievals make comparison harder. On a monthly scale, Table 3 indicates that all models where comparison is possible agree with the mean September and October observed heights within error, providing confidence that the models adequately capture the plume height within the Holuhraun region at this temporal resolution.

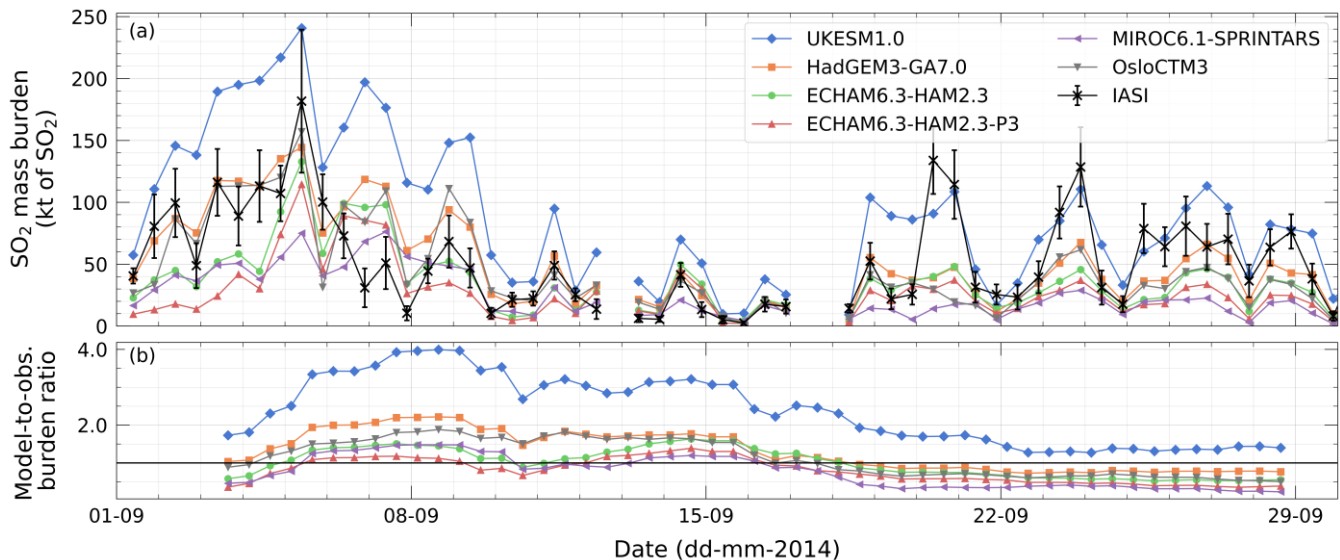

**Figure 4:** (a) Bidaily temporal evolution and (b) 5-day rolling mean of IASI retrieved and modelled SO$_2$ mass burdens across
the Holuhraun region (44°N to 80°N, 60°W to 30°E) for September 2014. Black crosses and error bars represent the mass
burdens and associated error derived from the IASI retrievals. Models are sampled only within the observed plume extent and
their derived mass burdens are given by the coloured lines. The 5-day rolling means tolerate a maximum of two missing data
points. Both IASI and model data have been regridded onto the same 1.0° x 1.0° latitude-longitude grid. The same grid cells
used in Fig. 3 are used to derive the mass burdens shown here.

The IASI retrieved and modelled SO$_2$ mass burdens across the Holuhraun region for September and October 2014 are shown
in Fig. 4a and the Supplement (S4) respectively, with monthly values provided in Table 3. Both the observed and modelled
mass burdens are derived by summing the product of the SO$_2$ column load and surface area of the individual grid cells within
the observed plume extent for each bidaily interval. The same method is applied to the SO$_2$ column load IASI retrieval error
to estimate an SO$_2$ mass burden observational error. For September and October 2014, we estimate an average bidaily SO$_2$
mass burden of 52 ± 15 kt of SO$_2$ and 34 ± 9 kt of SO$_2$ from IASI retrievals respectively which is in excellent agreement with
Malavelle et al. (2017) who report corresponding mass burdens of 52 kt of SO$_2$ and 30 kt of SO$_2$ using an independent IASI
dataset for the same geographical region. September mass burdens derived in Schmidt et al. (2015) using the Ozone Monitoring
Instrument (OMI) are considerably higher, averaging 99 ± 49 kt of SO$_2$, across a slightly smaller area (45°N to 75°N, 60°W to
30°E). There exists substantial bidaily variation in the observed mass burden evident by the peaks of 180 ± 60 kt of SO$_2$, 130
± 30 kt of SO$_2$, and 130 ± 30 kt of SO$_2$ on 5[th], 20[th], and 23[rd] September respectively, and the low values below 15 kt of SO$_2$
(e.g. 13[th]–16[th] September). This variation is likely caused by a combination of the plume passing in and out of the defined
region, changing IASI retrieval coverage (see Fig. 3c), and fluctuations in the volcanic SO$_2$ emission flux (Thordarson and
Hartley, 2015).


With respect to the models, HadGEM3-GA7.0, ECHAM6.3-HAM2.3, and OsloCTM3 simulate average bidaily $SO_2$ mass burdens that lie close to those of the IASI retrievals for September and October 2014. UKESM1.0 overestimates the observed mass burdens, particularly during the early stages of September, which is potentially due to overpredicting total column $SO_2$; a bias that has been noted previously (Hardacre et al., 2021). As the IASI instrumentation is not able to sample the full

intricacies of the plume, the observed $SO_2$ mass burden presented here is to be considered as a lower estimate and so UKESM1.0 exceeding this total may not necessarily be an indicator of poor performance. ECHAM6.3-HAM2.3-P3 and MIROC6.1-SPRINTARS largely underestimate the IASI derived mass burdens, yet as the models are only sampled within the observed plume extent, these two models, as well as the remaining models, may simulate considerable mass in regions outside this extent (hatched areas in Fig. 2 and animation, S2). All models capture the observed variability, simulating larger mass

burdens during early September when the eruption is most powerful and prescribed emission rates the highest, before decreasing during October. Correcting the IASI retrievals for parts of the $SO_2$ plume potentially missing has proved valuable (e.g. Carboni et al., 2019a) and could improve the comparison of the modelled heights and mass burdens presented here, yet as the general variability of both characteristics is well captured and no significant defects exist, using a cloud-adjusted correction is deemed unnecessary here.


**Table 3:** Monthly IASI retrieved and simulated $SO_2$ plume heights and $SO_2$ mass burdens across the Holuhraun region (44°N to 80°N, 60°W to 30°E) for September (S) and October (O) 2014. Plume heights for MIROC6.1-SPRINTARS and ECHAM6.3-HAM2.3-P3 are derived from monthly resolution sampling, rather than bidaily as is done for other model estimates. The OsloCTM3 simulation does not contain the required diagnostics for a plume height estimate.


| | | | IASI | UKESM1.0 | HadGEM3-GA7.0 | MIROC6.1-SPRINTARS | ECHAM6.3-HAM2.3-P3 | ECHAM6.3-HAM2.3 | OsloCTM3 |
|---|---|---|---|---|---|---|---|---|---|
| $SO_2$ plume height (km AMSL) | Mean | S | 1.2 ± 0.5 | 1.8 ± 0.3 | 1.6 ± 0.3 | 1.1 ± 0.5 | 1.8 ± 0.5 | 2.1 ± 0.5 | - |
| | | O | 1.7 ± 0.7 | 2.3 ± 0.3 | 2.2 ± 0.3 | 1.5 ± 0.5 | 2.1 ± 0.5 | 2.7 ± 0.5 | - |
| | Max. | S | 3.6 ± 0.9 | 3.8 ± 0.3 | 2.7 ± 0.3 | 2.8 ± 0.5 | 2.5 ± 0.5 | 3.6 ± 0.5 | - |
| | | O | 3.2 ± 1.5 | 4.2 ± 0.3 | 4.8 ± 0.3 | 2.7 ± 0.5 | 2.8 ± 0.5 | 4.1 ± 0.5 | - |
| | Min. | S | 0.33 ± 0.16 | 0.7 ± 0.3 | 0.7 ± 0.3 | 0.3 ± 0.5 | 0.5 ± 0.5 | 1.0 ± 0.5 | - |
| | | O | 0.20 ± 0.13 | 0.6 ± 0.3 | 0.6 ± 0.3 | 0.5 ± 0.5 | 0.7 ± 0.5 | 1.4 ± 0.5 | - |
| $SO_2$ mass burden (kt of $SO_2$) | Mean | S | 52 ± 15 | 88 | 51 | 25 | 27 | 36 | 44 |
| | | O | 34 ± 9 | 45 | 26 | 13 | 22 | 30 | 23 |
| | Max. | S | 180 ± 60 | 241 | 144 | 76 | 114 | 133 | 157 |
| | | O | 230 ± 20 | 112 | 68 | 37 | 61 | 76 | 57 |
| | Min. | S | 2.6 ± 1.1 | 9.2 | 2.9 | 1.7 | 2.1 | 4.3 | 1.2 |
| | | O | 1.8 ± 1.1 | 2.2 | 1.4 | 0.1 | 0.6 | 1.9 | 0.9 |

# 5 Sulphurous Surface Mass Concentrations

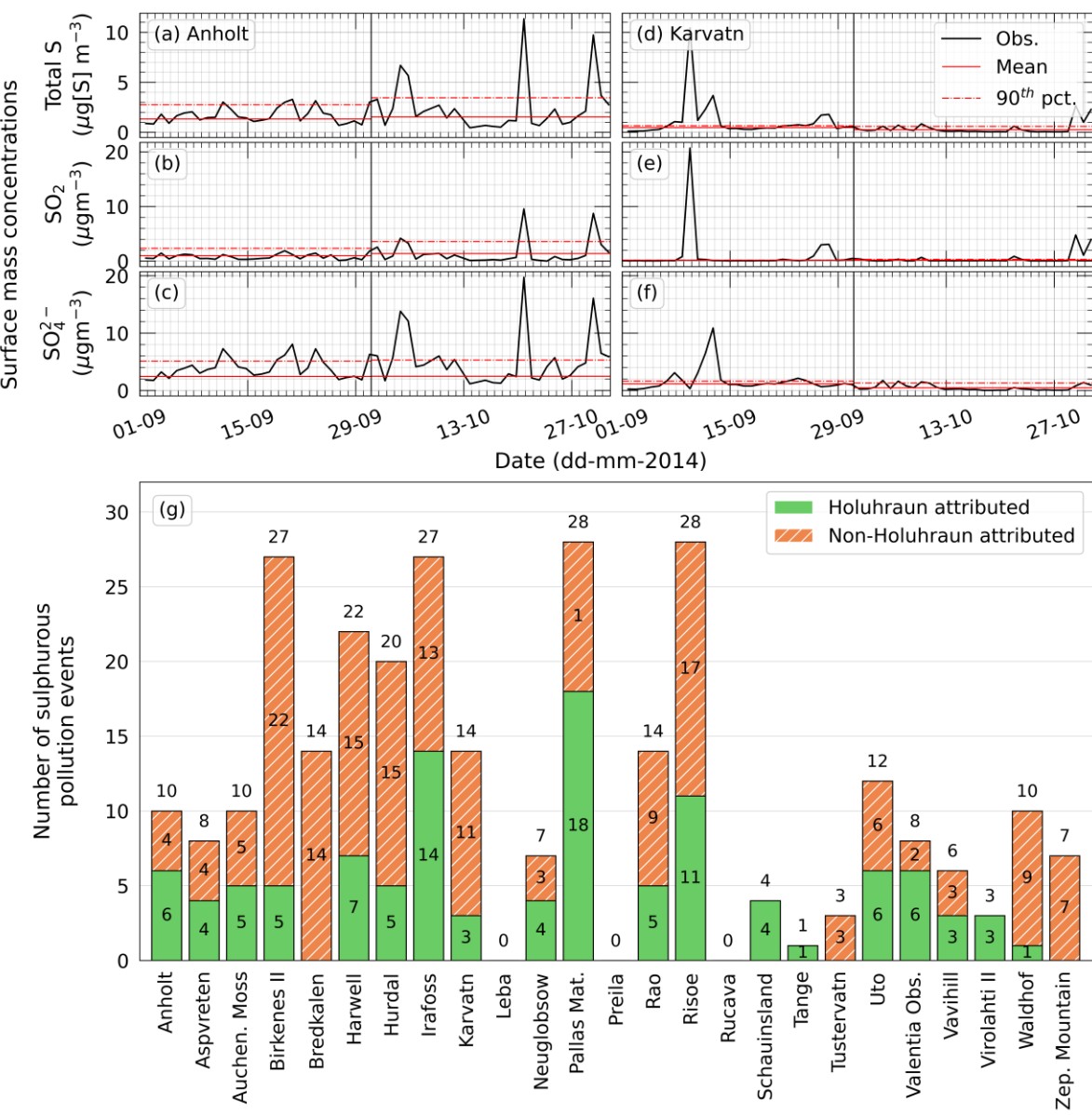

**Figure 5:** Observed surface mass concentration time series of total sulphur content, $SO_2$ and $SO_4^{2-}$ at (a, b, c) Anholt (Denmark) and (d, e, f) Kårvatn (Norway) between 1st September and 31st October 2014. Climatological monthly mean and 90th percentile values are given in the red solid and dashed lines respectively. (g) Sulphurous pollution events (see Sect. 2.2) identified across the 25 EMEP stations highlighting those attributed to Holuhraun emissions (green solid fill) and those not (orange hashed fill).

Observed time series of surface mass concentrations of total sulphur content, $SO_2$ and $SO_4^{2-}$ from 1$^{st}$ September to 31$^{st}$ October

2014 at EMEP stations Anholt (Denmark) and Kårvatn (Norway) are shown in Fig. 5a-c and Fig. 5d-f respectively. Time series of the remaining EMEP stations are provided in the Supplement (S5-27). Anholt and Kårvatn feature numerous peaks in sulphurous concentrations that exceed the climatological monthly statistics suggesting that these concentrations are significant and, given the rural locations of the sites, are likely caused by far afield sources of pollutants. Using the definition given in Sect. 2.2, we find that Anholt experienced 10 pollution events during September and October whilst Kårvatn experienced 14.

We see merit in defining a pollution event using the total sulphur content concentration, rather than the commonly used $SO_2$ concentration (e.g. Boichu et al., 2019), as additional events are identified due to their high $SO_4^{2-}$ concentrations which otherwise would have been missed (e.g. 20$^{th}$ and 23$^{rd}$ September at Anholt, 12$^{th}$ September at Kårvatn). The number of sulphurous pollution events observed across the 25 EMEP stations during September and October 2014 is shown in Fig. 5g. Birkenes II (Norway), Irafoss (Iceland), Pallas Matorova (Finland) and Risoe (Denmark) all experienced roughly an event

every two days, whilst only Leba (Poland), Preila (Lithuania) and Rucava (Latvia) did not experience any pollution episodes. In total, 283 pollution events are observed at 22 EMEP stations indicating that widespread sulphurous pollution occurred across Europe in the months following the eruption.

The likelihood of Holuhraun being a main source of pollution for the 283 events can be established qualitatively using the

IASI retrieved and modelled $SO_2$ column load animations or more robustly using the trajectory framework outlined in Sect. 2.4. Using the latter approach, the main source of pollution for 111 (39.2%) of the events can be attributed to Holuhraun emissions (see Fig. 5g for a station-by-station breakdown). Of the 22 EMEP stations experiencing a sulphurous pollution episode between September and October 2014, 19 stations endured at least one event influenced by the eruption. Note that other sources of pollutants may contribute to the mass concentrations observed at these 111 events, yet these contributions are

likely minor given the rural setting of EMEP stations and that Holuhraun is the dominant sulphurous source in the region covering this period. None of the combined 17 events observed at Bredkälen (Sweden) and Tustervatn (Norway) are attributed to Holuhraun emissions which, given that the plume has been shown to pass this area (e.g. Grahn et al., 2015; Ialongo et al., 2015), suggests an inconsistency in the trajectory analysis. This inconsistency could be resolved by revising the heights the trajectories are released at, by incorporating additional meteorological datasets and/or trajectory models, or by using a more

comprehensive trajectory framework (e.g. dispersion modelling). As assessing our trajectory framework is not the focus here, these inconsistencies have not been explored further. Nevertheless, the trajectory analysis shows that Holuhraun brought about significantly elevated sulphurous surface mass concentrations across Europe in September and October 2014; a testament to the sheer volume of $SO_2$ emitted into the region by the eruption.

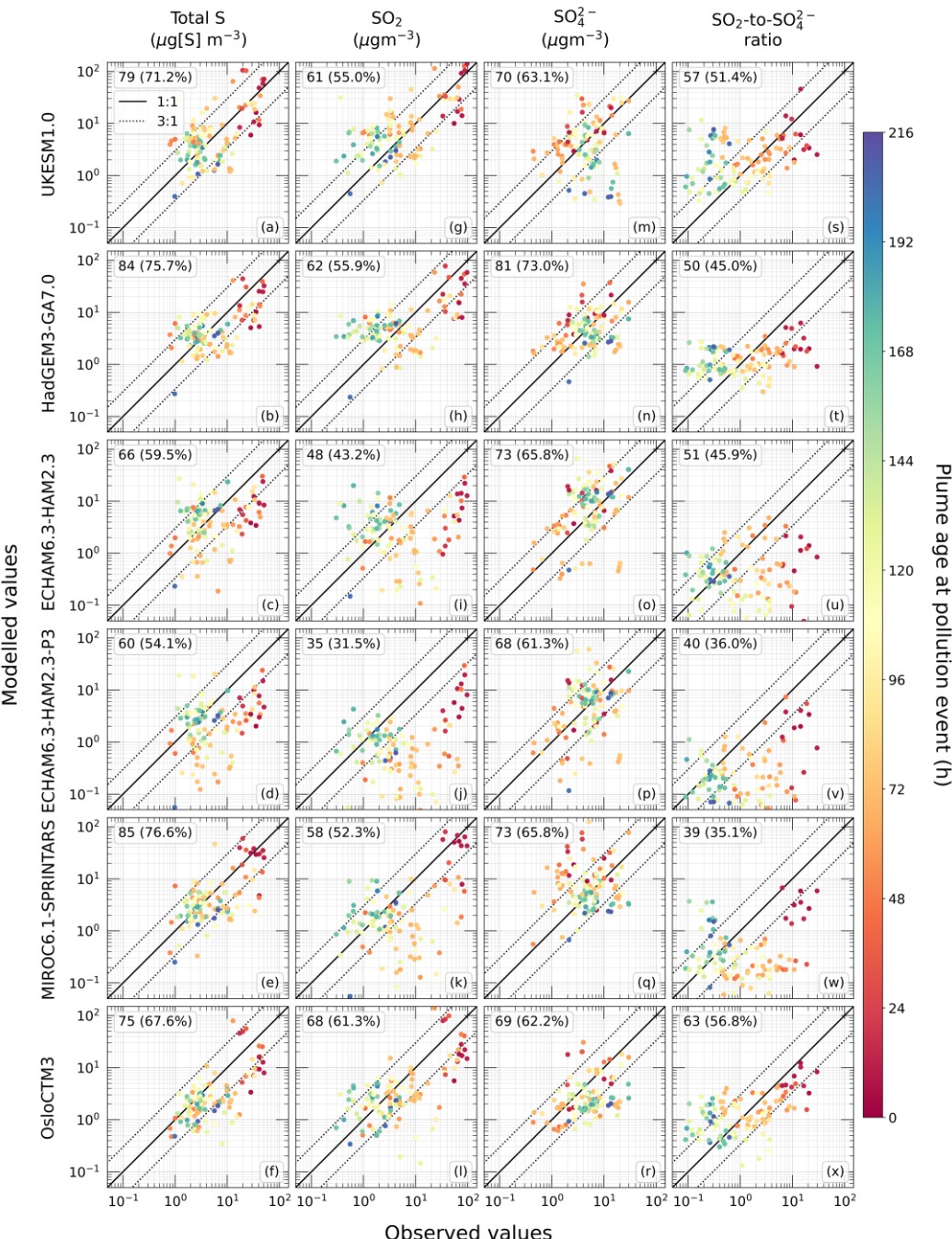

**Figure 6:** Modelled versus observed surface mass concentrations of total sulphur, $SO_2$ and $SO_4^{2-}$, and the $SO_2$-to-$SO_4^{2-}$ ratio for sulphurous pollution events attributed to Holuhraun emissions across the EMEP network for September and October 2014. Black solid and dashed lines represent parity and the 3-to-1 region respectively, with counts of points within the latter given. Colouring illustrates the plume age at the time of sampling. Observational errors are too small to discern.

Our catalogue of 111 sulphurous pollution events attributed to Holuhraun emissions is used to assess model performance in capturing the surface level behaviour of the plume. Fig. 6 displays the modelled versus observed surface mass concentrations of total sulphur (a-f), $SO_2$ (g-l), and $SO_4^{2-}$ (m-r), and the $SO_2$-to-$SO_4^{2-}$ ratio (s-x) of the volcanically influenced episodes. Colouring is used to highlight the plume age at the time of sampling (see Sect. 2.4). In terms of reproducing the observed total sulphur content, MIROC6.1-SPRINTARS, HadGEM3-GA7.0, and UKESM1.0 perform well with 76.6%, 75.7%, and 71.2%

of values within the 3-to-1 range respectively, whilst the two ECHAM variants capture just below 60%. These lower values are largely due to underestimating the higher sulphur content observed in the young plume (0–96 h) which itself is due to a considerable underestimation of the underlying $SO_2$ during these early stages of the plume. The $SO_2$ surface concentrations are also underestimated in MIROC6.1-SPRINTARS, yet for a slightly more mature plume (72–120 h), whilst the opposite is evident in UKESM1.0 and HadGEM3-GA7.0, predominantly for plume ages exceeding 96 h. The overestimation of European

surface $SO_2$ concentrations in UKESM1.0 has been noted previously (Hardacre et al., 2021), albeit over longer timescales. OsloCTM3 simulates the $SO_2$ well, showing no apparent overprediction or underprediction. All models depict the observed decrease in $SO_2$ concentration with increasing plume age.

With respect to $SO_4^{2-}$, all models improve on their $SO_2$ performance with each model having at least 61% of simulated

concentrations within the 3-to-1 range. The models show no obvious overestimation or underestimation, aside from UKESM1.0 and MIROC6.1-SPTRINARS where concentrations for plume ages above and below 96 h are generally underpredicted and overpredicted respectively. This underprediction of surface $SO_4^{2-}$ across Europe by UKESM1.0 has been stated previously (Hardacre et al., 2021; Mulcahy et al., 2020). In addition, the models struggle most in capturing the observed $SO_2$-to-$SO_4^{2-}$ ratios with only OsloCTM3 and UKESM1.0 having more than 50% of simulated values within 3-to-1 of the

observed. Broadly, all models, except for the ECHAM variants, simulate lower ratios than observed for the young plume (0–96 h) and higher ratios for the mature plume (144–216 h). The two ECHAM variants largely underestimate the observed ratio across all plume ages. There exists a notable underprediction in the ratio by MIROC6.1-SPTRINARS for plume ages roughly between 30 h and 60 h. Both the observed and modelled ratios decrease with increasing plume age suggesting that $SO_2$ oxidation to $SO_4^{2-}$ is occurring within the observed and simulated plumes.


Possible causes of differences between observed and simulated surface level behaviour of far afield Holuhraun pollutants, such as vertical resolution, source emission profile, and sub grid turbulence parameterisations, have been explored in depth previously (e.g. Boichu et al., 2016; Schmidt et al., 2015) and so will not be explored further here. As these challenges are not specific to Holuhraun and feature extensively in most numerical dispersion problems, the discrepancies that arise from them

should not act as evidence against the use of these models for the ACI investigation in Part 2 of this study. In fact, given the relatively fine spatial and temporal resolution that these coarse models are being assessed against here, they perform commendably in capturing the surface level behaviour of the plume.

## 6 In-Plume SO$_2$ oxidation to SO$_4^{2-}$

We have demonstrated that Holuhraun emissions affected the troposphere over long distances triggering SO$_2$ and SO$_4^{2-}$ pollution events across Europe and that the ratio of SO$_2$-to-SO$_4^{2-}$ during these episodes decreases as the plume ages. This suggests that SO$_2$ oxidation to SO$_4^{2-}$ is occurring as the plume matures. There are two main pathways for this conversion in the troposphere: gas-phase reactions, largely with the hydroxyl radical (OH-), and aqueous-phase reactions with dissolved ozone (O$_3$) and hydrogen peroxide (H$_2$O$_2$) (e.g. Calvert et al., 1978; Stevenson et al., 2003). The ratio of SO$_2$-to-SO$_4^{2-}$ is therefore useful in assessing whether oxidation processes are being accurately represented in the models; a ratio greater than that observed suggests that the overall oxidation processes are too slow whilst a ratio less than that observed suggests that the overall oxidation processes are too fast. This assessment is carried out here using the 111 pollution events attributed to Holuhraun emissions. By focusing on the ratio of the two pollutants as opposed to the total sulphur content and assuming that volcanic SO$_2$ and SO$_4^{2-}$ coexist, the variation in the absolute Holuhraun daily sulphurous emission flux can be ignored.

Fig. 7a shows the observed surface mass concentration ratio of SO$_2$-to-SO$_4^{2-}$ on a logarithmic scale of the 111 sulphurous pollution events attributed to Holuhraun emissions versus the age of the plume at the time of sampling, with colouring highlighting the plume's travel distance from the eruption (see Sect. 2.4). A variety of plume ages and SO$_2$-to-SO$_4^{2-}$ ratios make it apparent that the plume is sampled during different stages of maturity. The linear characteristics of Fig. 7a, along with the curve depicted in the equivalent linear scale figure in the Supplement (S28), imply an exponential decay of the SO$_2$-to-SO$_4^{2-}$ ratio with plume age; a relationship commonly associated with the depletion of volcanic SO$_2$ and one that assumes first order kinetics with respect to the SO$_2$ concentrations (e.g. Ilyinskaya et al., 2021; McGonigle et al., 2004; Oppenheimer et al., 1998; Pattantyus et al., 2018). Consequently, here we fit the observed data to an exponential decay function using non-linear total least squares regression as this considers both the ratio and plume age uncertainties. The derived SO$_2$-to-SO$_4^{2-}$ oxidation rate constant is $0.032 \pm 0.002$ h$^{-1}$ corresponding to an $e$-folding time of $1.30 \pm 0.08$ days. Using IASI retrieved SO$_2$ column load, Carboni et al. (2019a) estimate Holuhraun SO$_2$ depletion as having a mean 6-month $e$-folding time of $2.4 \pm 0.6$ days, whilst Schmidt et al. (2015) derive a mean September SO$_2$ $e$-folding time of $2.0 \pm 0.8$ days using NAME simulations of the eruption. Whilst not directly comparable, as these studies have not estimated the oxidation rate explicitly and focus on different time periods, both estimates are of similar magnitude to the SO$_2$ oxidation $e$-folding time found here. Assuming our exponential decay relationship holds close to the eruption vent, this study estimates a near-vent SO$_2$-to-SO$_4^{2-}$ ratio of $25 \pm 5$. This result agrees with Ilyinskaya et al. (2017) who report SO$_2$-to-SO$_4^{2-}$ ratios of 2 to 250 and 4 to 94 at 100 km and 250 km from the vent respectively. Boichu et al. (2019) estimate a slightly lower near-vent SO$_2$-to-SO$_4^{2-}$ ratio of 19.7 using a linear model created from 5 observations to describe the evolution of the SO$_2$-to-SO$_4^{2-}$ ratio.

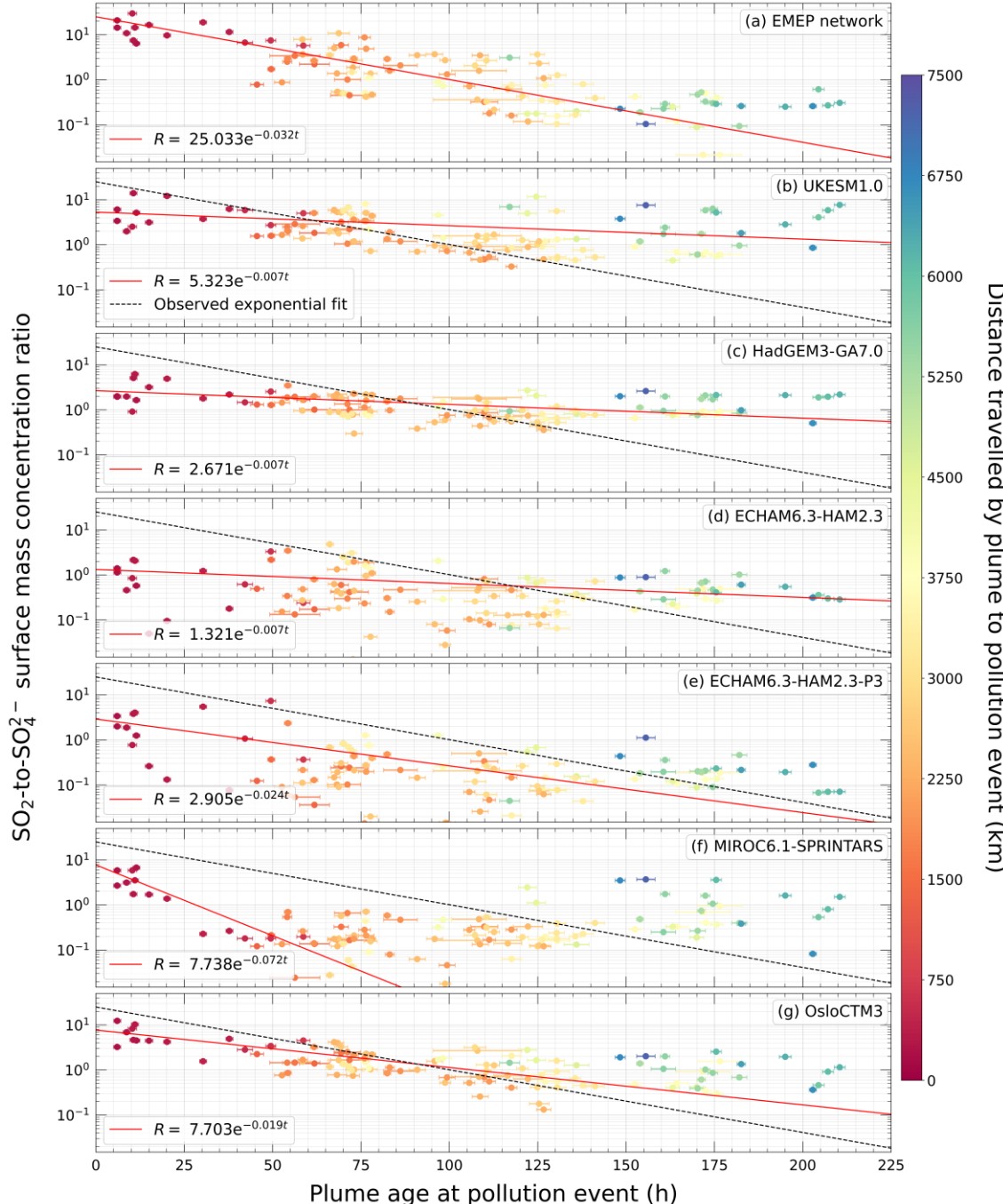

**Figure 7:** (a) Observed and (b-g) modelled $SO_2$-to-$SO_4^{2-}$ surface mass concentration ratios of sulphurous pollution events attributed to Holuhraun emissions with respect to the plume age at the time of sampling for September and October 2014. Red lines represent the exponential decay fits. Black dashed lines are the observed exponential fit overlayed onto the modelled ratios. Colouring illustrates the plume's travel distance at the time of sampling Observational errors are too small to discern.

**Table 4:** Summary of the observed and modelled in-plume $SO_2$ oxidation to $SO_4^{2-}$ using the sulphurous pollution events attributed to Holuhraun emissions for September and October 2014.


|  | EMEP network | UKESM1.0 | HadGEM3-GA7.0 | MIROC6.1-SPRINTARS | ECHAM6.3-HAM2.3-P3 | ECHAM6.3-HAM2.3 | OsloCTM3 |
|---|---|---|---|---|---|---|---|
| Near-vent $SO_2$-to-$SO_4^{2-}$ ratio | $25 \pm 5$ | $5.3 \pm 0.8$ | $2.7 \pm 0.3$ | $7.7 \pm 1.5$ | $2.9 \pm 0.5$ | $1.3 \pm 0.2$ | $7.7 \pm 0.6$ |
| $SO_2$ oxidation rate constant ($h^{-1}$) | $0.032 \pm 0.002$ | $0.0069 \pm 0.0018$ | $0.0070 \pm 0.0012$ | $0.072 \pm 0.018$ | $0.024 \pm 0.004$ | $0.007 \pm 0.002$ | $0.0191 \pm 0.0016$ |
| $SO_2$ oxidation $e$-folding time (days) | $1.30 \pm 0.08$ | $6.0 \pm 1.6$ | $5.9 \pm 1.0$ | $0.58 \pm 0.15$ | $1.7 \pm 0.3$ | $5.8 \pm 1.8$ | $2.18 \pm 0.18$ |

Fig. 7b-g depict the modelled surface mass concentration ratio of $SO_2$-to-$SO_4^{2-}$ versus the age of the plume that these ratios are sampled in. All models display an exponential relationship between the $SO_2$-to-$SO_4^{2-}$ and plume age, albeit only for the young plume ($< 70$ h) in the case of MIROC6.1-SPRINTARS. Each model is fitted to an exponential decay function which is
given by the solid red line with the observed fit overlayed in the black dashed line for comparison. Due to how the plume age error increases with plume age, and how the magnitude of the ratio error is negligible in comparison, ratios sampled in the mature plume have a larger total error and so are weighted less in the fitting than those sampled in the younger plume. The exponential decay parameter estimates for the models are displayed in Table 4. The modelled near-vent ratios are all smaller than that derived from observations, yet still agree with those found in Ilyinskaya et al. (2017). Except for MIROC6.1-
SPRINTARS, all model derived oxidation rates are slower than that derived from observations, ranging from being roughly just under twice as slow in ECHAM6.3-HAM2.3-P3 and OsloCTM3, to 4.5 times as slow in UKESM1.0, HadGEM3-GA7.0, and ECHAM6.3-HAM2.3. The seemingly poor fit of MIROC6.1-SPRINTARS is due to the underestimation of the ratios between approximately 30 h and 60 h (Fig. 6w) which, for reasons stated previously, have a larger influence on the fit than ratios sampled in the more mature plume. If ratios across this time range were better captured by MIROC6.1-SPRINTARS, a
fit that results in a slower derived oxidation rate more in keeping with the other models and observations could be expected. Interestingly, there is no apparent correlation between a model's vertical resolution and a model's ability to capture the in-plume $SO_2$-to-$SO_4^{2-}$ oxidation.

## 7 Summary and Conclusions

By releasing 9.6–11.8 Mt of $SO_2$ into the lower troposphere across nearly 6 months, the 2014–2015 Holuhraun eruption offers an opportunity to challenge the capability of GCMs in capturing the characteristics of tropospheric sulphate aerosol intricacies resulting from effusive eruptions and assess the potential impact of subsequent aerosol-cloud interactions. A model inter-comparison effort has been initiated to leverage this opportunity and the results from Part 1 of the two-part analysis are presented here. Remote sensing data of $SO_2$, and surface level $SO_2$ and $SO_4^{2-}$ mass concentration measurements are used, in

conjunction with trajectory modelling, to evaluate the performance of 5 GCMs and a CTM in simulating the spatial and chemical evolution of the $SO_2$ plume across the North Atlantic and Europe.

A comparison against IASI $SO_2$ retrievals shows that the models capture the evolution of the volcanic plume within the surrounding region well during September and October 2014. Holuhraun emissions are the dominant source of $SO_2$ in the

models and the spatial transport of the associated $SO_2$ plume is well replicated. The $SO_2$ plume height is slightly overestimated by the models, whereas there is no general overestimation or underestimation in simulating the $SO_2$ mass burdens; it is model dependent. The temporal variability of both these plume characteristics is well captured. Discrepancies with the IASI retrievals could be due to several factors including the limitations of the IASI retrievals (e.g. Carboni et al., 2019a), and discrepancies between the idealised volcanic emission profile used by the models and the real emissions (e.g. Steensen et al., 2016). A

comparison against retrievals of volcanic $SO_2$ from other satellite instrumentations may yield different conclusions, yet the descriptions of the plume spatial distribution made with other remote sensing products are similar (e.g. OMI: Ialongo et al., 2015; Schmidt et al., 2015, Steensen et al., 2016; OMPS-NN: Ialongo et al., 2015; GOME-2: Twigg et al., 2016). Even though the model spatial representations of the eruption are not perfect, our intent here is rather to identify the variations in the models' transport of Holuhraun $SO_2$ as this will help discern the impact on cloud properties and assess the ACIs in Part 2.


By combining the surface mass concentration measurements of $SO_2$ and $SO_4^{2-}$ made during September and October 2014 across the EMEP network with single-particle trajectories calculated using the HYSPLIT model, the simulated surface level behaviour of the plume was assessed. Of the 283 sulphurous pollution events identified, 111 are attributed to Holuhraun emissions. Generally, the models reproduce the measured elevated surface level concentrations during these volcanically

influenced events, yet they struggle in simulating the correct magnitude, notably the ratio of $SO_2$-to-$SO_4^{2-}$ which is often underestimated and overestimated for the young and mature plume respectively. Although this should not be discouraging as capturing volcanic sulphurous pollutant surface mass concentrations far afield at a specific location and time is challenging even for CTMs of finer scales. We note that the models with finer vertical resolutions, UKESM1.0, HadGEM3-GA7.0, and OsloCMT3, describe ground-level concentrations of Holuhraun pollution episodes best; a feature that has been noted

previously by Boichu et al. (2016). Given the relatively coarse scale of the simulations discussed here, the surface level performance of the models is admirable.

Both the observed and modelled ratios of $SO_2$-to-$SO_4^{2-}$ surface mass concentrations sampled within the plume are shown to decrease with increasing time and distance from the eruption vent suggesting $SO_2$ oxidation to $SO_4^{2-}$ is occurring. To explore this further, the ratios as a function of plume age have been analysed revealing an exponential decay. By fitting this decay to an exponential function, observed and modelled rate constants for the volcanic $SO_2$ oxidation are estimated. Aside from MIROC6.1-SPRINTARS, the in-plume $SO_2$-to-$SO_4^{2-}$ oxidation is shown to be slower in the models than observed. This implies that the volcanic $SO_2$ introduced into the simulations may not be chemically converted fast enough relative to what is derived from surface measurements (if deposition effects are ignored). The considerable underestimation of the ratios sampled at plume ages between 30 h and 60 h is the suggested reasoning for MIROC6.1-SPRINTARS exhibiting opposing behaviour. No correlation between a model's vertical resolution and a model's derived $SO_2$ oxidation rate is found.

The oxidation rate constants explored here are generalised values representing both the gas-phase and aqueous-phase pathways. This study attempted to help elucidate the complexity of volcanic $SO_2$ oxidation by fitting the $SO_2$-to-$SO_4^{2-}$ ratios to a biexponential function, a sum of two individual exponential decay components, to distinguish between the two pathways by estimating individual gaseous and aqueous oxidation rate constants. Despite the success of previous studies in estimating and applying multiple rate constants to describe the depletion of volcanic $SO_2$ (Ilyinskaya et al., 2021; Pattantyus et al., 2018), this study found no significant improvement in the fitting function versus a standard exponential decay (one exponential decay component). Categorising the $SO_2$-to-$SO_4^{2-}$ ratios in terms of the conditions the air parcels are subject to during transport, such as time spent in-cloud, relative humidity, cloud pH, oxidant concentrations, time of day, and deposition rates, could reveal the dominant oxidation pathway/s affecting a particular group. The subsequent fitting versus plume age would then provide rate constant estimates of the mechanism/s in play. However, such a method would likely require a sophisticated Lagrangian framework, and it is beyond the scope of this work to explore the intricate chemical kinetics of volcanic $SO_2$ oxidation.

Overall, the 6 models considered here provide reasonable simulations of the spatial and chemical evolution of the Holuhraun plume and are considered competent enough to be used to explore the impacts of the eruption on ACIs in the region (see Part 2 of this study). It is important to acknowledge, and is possibly relevant to the wider ACI community, that this analysis has also highlighted that the models do not perfectly capture the secondary $SO_4^{2-}$ aerosol production during a large degassing event. This pitfall may contribute to the well-documented disagreements between model ACIs estimates as the underlying aerosol perturbations would differ. We hope that our application of in situ sulphurous surface measurements to assess numerical models helps bolster the case to retain and extend air monitoring networks of volcanic pollutants for use in future studies.

## Code Availability

Code is available from the corresponding author on reasonable request.

## Data Availability

The IASI $SO_2$ retrieval dataset is available on the CEDA Archive, https://catalogue.ceda.ac.uk/uuid/d40bf62899014582a72d24154a94d8e2 (Carboni et al., 2019b). The EMEP network surface $SO_2$ and $SO_4^{2-}$ mass concentrations are available through the EBAS database, https://ebas.nilu.no/data-access/. All model data, including trajectory output, used in this study is available on Zenodo, https://doi.org/10.5281/zenodo.10160538 (Jordan, 2023).

## Author Contributions

GJ, JH, and FM designed the experiment. GJ handled the remote sensing data and in situ surface measurements whilst GJ, DW-P, TT, DN, GM, and RS ran the model simulations. PK, DGP, and ED provided HYSPLIT trajectories and offered guidance in their use. GJ, JH, FM, YC, AP, ED, and DGP analysed the spatial and chemical evolution of the Holuhraun plume. GJ prepared the manuscript with contributions from all co-authors.

## Competing Interest

At least one of the (co-)authors is a member of the editorial board of Atmospheric Chemistry and Physics. The peer-review process was guided by an independent editor, and the authors also have no other competing interests to declare.

## Acknowledgements

GJ and JH were funded under the European Union's Horizon 2020 research and innovation programme under the CONSTRAIN grant agreement 820829. GJ, JH, and FM are supported by the Met Office Hadley Centre Climate Programme funded by DSIT. JH, YC, and AP would like to acknowledge funding from the NERC ADVANCE grant (NE/S015671/1). DGP would like to express his gratitude to Dr. Zak Kipling for providing support in obtaining HYSPLIT input files from ERA-Interim reanalysis data.

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
