# Peer review of "How well are aerosol-cloud interactions represented in climate models? Part 1: Understanding the sulphate aerosol production from the 2014-15 Holuhraun eruption."

_EGUsphere, 2023_

## Author Comment (AC1)

We would like to thank Andreas Stohl for taking the time to read and provide comments on our paper. We believe the changes made in response to their comments have strengthened the manuscript.

See below for our response (in red) to their comments (in black).

**Response to Andreas Stohl**

This paper describes an interesting analysis of the SO2-to-sulfate oxidation in models relative to those inferred from observations after the Holuhraun eruption. The paper shows that gas-phase oxidation rates in the models are all slower than the observed rates, which is an important result. The main result of the study is presented in Figure 6. However, I have a few concerns about this figure, as detailed below in my major comments below. Most importantly, I am not convinced that a robust separation between gas- and aqueous-phase oxidation is possible based on the available observation data, mostly for two reasons: 1) the mono- and bi-exponential fits are very similar, and it is not so clear that the bi-exponential fit is SIGNIFICANTLY better than the mono-exponential fit; 2) the attribution of the two e-folding times obtained by the fit to gas- and aqueous-phase oxidation seems quite a stretch. I think this interpretation needs independent support before the paper can be published. A few other points also need to be addressed, as outlined below.

Major:

The trajectory analysis is somewhat problematic. First of all, how are the 27 members of the trajectory ensembles (line 174) different from each other? This is not explained in the text. Second, all EMEP stations are located in the atmospheric boundary layer, where air mass trajectories are not well representing the properties of the flow, due to turbulence. This will likely affect the quality of the attribution of events to Holuhraun (or not). Third, the definition of "vicinity" of the Holuhraun eruption is highly subjective. Depending, e.g., on the transport time and distance, trajectory errors will likely be very much case-dependent, and a single "vicinity area" might not be appropriate for all cases (e.g., stations closer to Holuhraun will have a greater chance of hitting the defined vicinity area.

Thank you for raising your concerns regarding the trajectory analysis. We hope the following changes reassure you that our methodology is sound.

Firstly, a more detailed description on the trajectory analysis has been provided. This includes a clarification on how the 27 ensemble members differ due to small perturbations in the input meteorology data (i.e. offset by a fixed grid factor, a maximum of 1.0° of latitude/longitude in the horizontal and 0.01 sigma units in the vertical, and so all possible offsets result in the 27 members).

Secondly, we acknowledge there is often greater uncertainty in using single-particle trajectories as opposed to dispersion modelling. Nevertheless, single-particle trajectory frameworks like ours have been widely used in many previous studies to characterise long-range transport (e.g. Nieminen et al., 2015; Räty at al., 2023; Väänänen et al., 2013). Plus, our goal is to inter-compare models consistently rather than through a rigorous dispersion exercise, and so believe our trajectory framework is sufficient to achieve this. We have added additional comments to the manuscript discussing this.

In addition, the reviewer is right in that these uncertainties may affect the quality of attribution. However, the Holuhraun emissions were so substantial within the region that it would be extremely unlikely that most identified events were misattributed to Holuhraun. Moreover, sulphurous surface concentrations are rather low over Europe in recent times and spikes in $SO_2$ are a rare occurrence.

Finally, we agree that the "vicinity" of the Holuhraun eruption is subjective. To address this, we have taken a more quantitative approach. We now define multiple spherical bounding areas with radii increasing with a station's increasing distance from Holuhraun. These radii values are based on the positional error of a trajectory being approximately 10–30% of the total distance travelled (Stohl, 1998). A special case is made for the Irafoss station due to the typical spatial resolution of the trajectories being greater than the error estimated from 10-30% of the distance. Consequently, we define the bounding radius here using local wind speeds following the methodology set out in Hughes et al. 2012.

The comparison between models and IASI data is not fully convincing. It seems model output is shown irrespective of whether IASI retrievals are available for a location or not. IASI retrievals can easily miss volcanic SO2, e.g., underneath clouds. Thus, models should only be sampled in pixels where IASI SO2 retrievals are actually made. The authors write that models often have larger plume areas than the IASI retrievals, which can be attributed to clouds affecting IASI. Still, it appears that many models actually have often smaller plume areas than IASI. This would even be worse when cloud screening is applied.

We have listened to your suggestion and agree the better approach is to only sample models where successful IASI $SO_2$ retrievals have been made. Subsequently, our comparison with IASI retrievals is now only made on grid cells within the observed plume extent. On redoing the analysis, the modelled $SO_2$ plume heights now overestimate observations whilst no substantial changes in the $SO_2$ mass burden comparison is found.

On the comment "many models actually have often smaller plume areas than IASI", we acknowledge that this is true across certain periods, yet overall we observe that the modelled plume extents are larger than those observed, particularly in October. We have improved Fig. 2 and the animation by explicitly distinguishing between regions inside and outside the observed plume extent in the model simulations (coloured vs hatched areas) to improve the evidence for this statement.

Figure 5: Since the conversion rate of SO2 to sulfate is shown to be uncertain, I am wondering why Figure 5 does not also show a comparison for total sulphur (SO2 + sulfate). This should provide the most robust comparison between the models and the observations.

A total sulphur comparison has been added to the figure and the discussion extended.

Figure 6: This is the core result of the paper and quite interesting. However, I am not at all convinced that the bi-exponential fit is any better than the mono-exponential fit. That the bi-exponential fit is better (line 417) is a trivial result. But is it really SIGNIFICANTLY better? The two e-folding times obtained are interpreted as gas-phase and aqueous-phase e-folding times. But I am concerned that the fit is not stable enough to reliably distinguish between the two. Furthermore, how do you know which e-folding time is which? The data per se do not give any information on the two processes, but the authors immediately jump to the conclusion that these are gas- and aqueous phase e-folding times. What is the evidence for this?

We thank the reviewer for raising their concerns with the biexponential fitting. A similar comment was made from another reviewer. In hindsight, we acknowledge that attempting to separate the inplume SO$_2$ into its gaseous and aqueous pathways was too much of stretch for our dataset. We now only fit to an exponential with a single decay constant. The derived gaseous-phase and aqueous-phase oxidation rate constants are now replaced with a single value generalising them. However, we believe that our efforts to explore the complexity of volcanic SO$_2$ oxidation is still worth mentioning and a suggested direction for future works has been included.

Figure 6: The aqueous-phase oxidation occurs only in clouds, so is a single e-folding time even appropriate to characterize this oxidation? This must be highly variable, depending on the time the SO2 spends in a cloud.

See previous comment.

Figure 6: All events are exclusively attributed to Holuhraun. However, there are likely always (perhaps minor) contributions from other sources. How might these affect the results, especially far away from the volcano, where SO2/sulfate ratios are low and even relatively small anthropogenic SO2 emissions could affect the ratio substantially.

Our fitting considers both the uncertainty in the ratio and plume age, with the latter being by far the dominant source. Generally, the plume age error increases with increasing plume age. Consequently, ratios sampled in a mature plume have less influence on the fitting parameters than those sampled in a young plume. As you say, the ratios "far away from the volcano" are low and so possibly are affected by small anthropogenic emissions, whilst the ratios sampled closer to the eruption are larger and less likely to be significantly affected. However, these "far away" ratios have a limited influence on the fitting due to their associated large plume age errors, whereas the ratios with a low likelihood of anthropogenic impact have small plume age errors. Hence, the ratios that could be potentially affected by small anthropogenic SO$_2$ emissions substantially have a relatively minor influence on the overall oxidation rate constant and near-vent ratios derived.

In addition, a major goal of the EMEP network is to gather observations at locations where significant sources of local pollution are minimised (Tørseth et al., 2012). As such, anthropogenic SO$_2$ contributions in the observations used here should be minimal.

Line 435: A modelled event is considered successful if both SO2 and sulfate concentrations are within a factor 5 of the observations. Doesn't this introduce a bias in the analysis? You show that modelled oxidation rates are too slow – in this case one would expect the model to often substantially overestimate observed SO2 concentrations. But large overestimations would be substantially removed from the analysis, which would lead to biased results.

This is a good point and is seemingly a flaw in our analysis. As such, we now no longer only fit model output to the events captured within a factor of 5. All models are fitted to all observed events attributed to Holuhraun preventing this bias from impacting our results. The new analysis without the bias does not change remarkably.

Line 161 and Table 2: Why are ERA-Interim reanalyses used? These are superseded since quite a few years already by ERA5 reanalyses with better resolution, and which should have better quality!

The experiment was initialised prior to the public release of ERA5 (2019) which is why it is not included. As to why it has taken so long to reach the results stage, that is an unfortunate multitude of disruptions including COVID, a change of experiment lead, and submission errors.

Lines 279-280: How do you know that varying IASI SO2 burdens are due to changing IASI retrieval coverage and plumes passing in and out of the region, and not due to variations in emission flux? I don't think there is good enough data to prove that the emission flux was really constant.

We agree that there is not sufficient data to suggest the volcanic $SO_2$ emission flux was constant during September and October 2014 and that the coverage of IASI retrievals was too scarce to establish definitive conclusions. Rather, it is more likely variation existed (e.g. Thordarson and Hartley, 2015) and so contributes to the mass burden variation shown. Our comment has been amended to reflect this.

Lines 370-371: Why is a poor performance of concentration ratios expected? The two species are not simulated independently, so a plume in one species should always correlate with a plume in the other species.

Agreed, we have removed this comment.

Table 3: Why does OsloCTM3 not have the data required for filling Table 3? This should be basic model output (SO2 concentration fields) that is also needed for all other analyses?

The $SO_2$ columns loads and mass burdens for OsloCTM3 have now been included and the discussion amended accordingly. Unfortunately, from the diagnostics made available in the submission for this experiment it is not possible to derive an $SO_2$ plume height for OsloCTM3.

Line 55: word aerosol is duplicated in text

Corrected, thank you.

Hughes, E. J., Sparling, L. C., Carn, S. A., and Krueger, A. J.: Using horizontal transport characteristics to infer an emission height time series of volcanic $SO_2$, J. Geophys. Res., 117, D18307, doi:10.1029/2012JD017957, 2012.

Nieminen, T., Yli-Juuti, T., Manninen, H. E., Petäjä, T., Kerminen, V.-M., and Kulmala, M.: Technical note: New particle formation event forecasts during PEGASOS–Zeppelin Northern mission 2013 in Hyytiälä, Finland, Atmos. Chem. Phys., 15, 12385–12396, doi:10.5194/acp-15-12385-2015, 2015.

Räty, M., Sogacheva, L., Keskinen, H.-M., Kerminen, V.-M., Nieminen, T., Petäjä, T., Ezhova, E., and Kulmala, M.: Dynamics of aerosol, humidity, and clouds in air masses travelling over Fennoscandian boreal forests, Atmos. Chem. Phys., 23, 3779–3798, doi:10.5194/acp-23-3779-2023, 2023.

Stohl, A.: Computation, accuracy and applications of trajectories—A review and bibliography, Atmos. Environ., 32, 947-966, 1998.

Thordarson, T., and Hartley M.: Atmospheric sulfur loading by the ongoing Nornahraun eruption, North Iceland, Geophys. Res., Abstracts, 17(EGU2015-10708), 2015.

Tørseth, K., Aas, W., Breivik, K., Fjæraa, A. M., Fiebig, M., Hjellbrekke, A. G., Lund Myhre, C., Solberg, S., and Yttri, K. E.: Introduction to the European Monitoring and Evaluation Programme (EMEP) and observed atmospheric composition change during 1972–2009, Atmos. Chem. Phys., 12, 5447–5481, doi:10.5194/acp-12-5447-2012, 2012.

Väänänen, R., Kyrö, E.-M., Nieminen, T., Kivekäs, N., Junninen, H., Virkkula, A., Dal Maso, M., Lihavainen, H., Viisanen, Y., Svenningsson, B., Holst, T., Arneth, A., Aalto, P. P., Kulmala, M., and Kerminen, V.-M.: Analysis of particle size distribution changes between three measurement sites in northern Scandinavia, Atmos. Chem. Phys., 13, 11887–11903, doi:10.5194/acp-13-11887-2013, 2013.

---

## Author Comment (AC2)

We would like to thank the anonymous referee for taking the time to read and provide comments on our paper. We believe the changes made in response to their comments have strengthened the manuscript.

See below for our response (in red) to their comments (in black).

**Response to anonymous referee**

For two months of the Holuhraun fissure eruption in 2014 and 2015, this study presents an analysis of the sulphur and sulphate dispersion over Europe as well as an evaluation of the sulphate production. For this, dispersion simulations of multiple models are utilised, compared, and evaluated against IASI SO2 retrievals and surface concentration measurement of European monitoring stations. These comparisons are fairly conducted and discussed. However, to a large extend the authors seem to focus on model comparison and evaluation. Consequently, the manuscript could partly be more adequate to be published in GMD rather than ACP. However, the investigation on SO4 formation fits well with the purpose of ACP and should be emphasised more in the abstract. Furthermore, there are multiple aspects in the model intercomparison, the trajectory-based source assignment, and the SO2-to-SO4 conversion that need further elaboration before the manuscript may be published.

Major comments:

A multi-model intercomparison necessarily needs to consider differences in the models' characteristics as model physics/dynamics/chemistry and the model setup as parameterisations and model resolution to allow for a fair comparison and meaningful discussion. Therefore, I suggest adding the advection schemes used in Table 2. How can the OsloCTM3 not have a chemistry/aerosol module as a CTM? Furthermore, the vertical model level distribution is essential with respect to the volcanic emission plume. Please include the model layer thickness of the lowest model level and the number of model layers between 0 km and 3 km height in section 2.3 and consider this in the discussions! How do the different models differentiate with respect to vertical layering, vertical distribution of emissions etc.? Further, it is very hard to assess the performance of the different models individually, as the evaluation and the figures/tables often include just a selection of the models. It seems to be not a full and fair comparison between all models. Figure 2 does not include the CTM due to missing required diagnostics (lines 290-291). What does this mean? Why is it then being listed in table 3? Figure 3 just evaluates the performance of 3 models. Why are the model outputs not designed in the way that they are comparable? Please consider producing comparable model output to fully discuss all model performances.

Thank you for your suggestions on how to improve the model inter-comparison part of our study.

Firstly, you are correct in that the model vertical resolutions need to be considered more. Subsequently, details on the model vertical resolution have been added to Table 2 and considered during the discussions. However, we have not included the advection schemes. As the models are nudged to ERA-Interim reanalyses data, the dispersion is already constrained which was done, partly, to mitigate the transport errors. Hence, we feel added details on advection schemes is superfluous.

Secondly, you are right that OsloCTM3 has a chemistry scheme, and its omission is an error that has been corrected.

Moreover, all figures inter-comparing model output now contain all models where possible.

Finally, the SO$_2$ columns loads and mass burdens for OsloCTM3 have now been included and the discussion amended accordingly. Unfortunately, from the diagnostics made available in the submission for this experiment, it is only possible to derive monthly SO$_2$ plume height estimates for MIROC6.1-SPRINTARS and ECHAM6.3-HAM2.3-P3, whilst no estimate is possible for OsloCTM3. Whilst we agree with the reviewer that having all the diagnostics requested for the experiment would be ideal, the main aim of the experiment is the investigation of the aerosol-cloud interactions (ACIs) in Part 2. As a significant number of diagnostics were requested and the submissions made "pro-bono", some modelling centres focused their resources on the diagnostics required for the ACI study leaving some diagnostics absent for Part 1. We agree this limits the plume height inter-comparison, yet still believe the discussions on the output available to the experiment warrants inclusion.

The setup of the backward trajectories simulated with HYSPLIT needs clarification (lines 174-178). It remains unclear how the ensemble trajectories are designed. Are there 27 ensemble members defined for each station and each hour? If so, how are the ensemble members at the individual stations perturbed? Or do you have one ensemble member per hour for each station? However, 27 stations do not agree with table 1. Table 1 lists the starting heights for the trajectories at each station. How are these heights defined? Is there a basis for these heights as the station height above the surface height of the corresponding GCM's grid cell? How do you ensure that the trajectories allow for a fair comparison with ground-based observations? Further, are 2 % of the trajectories passing through the 3D bounding box (line 184) significant enough to connect these to the Holuhraun event?

Thank you for raising your concerns regarding the trajectory analysis. We hope the following changes reassure you that our methodology is sound.

Firstly, a more detailed description on the trajectory analysis has been provided. This includes a clarification on how the 27 ensemble members differ due to small perturbations in the input meteorology data (i.e. offset by a fixed grid factor, a maximum of 1.0° of latitude/longitude in the horizontal and 0.01 sigma units in the vertical, and so all possible offsets result in the 27 members).

Secondly, the trajectory starting heights for each station used offsets to the model surface to address differences between the orography in the driving meteorological data and reality.

Thirdly, we agree that the 2% threshold was too small to be confident in attributing a pollution event to Holuhraun emissions. The threshold has since been increased to 25%.

The assessment of the transport time (lines 185-192) is not fully clear and needs some rephrasing. I would have expected a circular influence region around the eruption site instead of a squared bounded domain. Please also elaborate on the definition of the idealised trajectory points in Fig. 1. Do these relate to full hours? And how many black circles are attributed to which trajectory?

We agree that a circular influence region is more suitable and have made changes to our trajectory framework to include such. In our revised framework, the idealised trajectory points in Fig. 1 are no longer relevant and have been removed.

Regarding the IASI SO2 retrieval, it remains unclear what "the SO2 detection is positive" means (lines 112-114). Is 0.49 DU a detection limit? Or is it an individually defined value to discriminate volcanic SO2 from other SO2 (which is the climatology)? Please clarify this. Furthermore, in line 118, you are referring to meteorological temperature profiles for the height conversion. Where do these profiles come from? Are they standard profiles or from meteorological model analyses? To better understand the uncertainty of the IASI retrievals, it would be desirable to have a short summary of the different components contributing to the retrieval error. For example, do the uncertainties of assuming Gaussian profiles and the uncertainties of the temperature profiles contribute to the retrieval error? With respect to the discussion in line 249, please justify why the central height of a Gaussian SO2 vertical profile can be an estimate for the injection height. There are enormous amounts of mass being distributed above the central height of a Gaussian profile.

Thank you for seeking additional clarity on the IASI SO$_2$ retrievals.

The 0.49 DU is a threshold set by Carboni et al., 2019 within their IASI retrieval algorithm and is chosen specifically for the Holuhraun eruption. The threshold is used in the detection scheme which is a linear retrieval with the SO$_2$ column load as the free parameter. The linear retrieval assumes the background SO$_2$ concentration follows a Gaussian distribution. The threshold is set substantially higher than the standard deviation of this distribution meaning a positive result is exceedingly likely to be significantly different to the background. We have improved the clarity of this section within the manuscript. Further details on the detection scheme can be found in Walker et al. 2011, 2012.

In terms of the height conversion, this is performed within the algorithm in Carboni et al., 2019 and is not a post processing step by us. They use atmospheric profiles from European Centre for Medium-Range Weather Forecasts (ECMWF) meteorological data.

We have added extra details on the IASI error estimates and provided references to seek further information. Although considered important, the reason for using the Carboni et al., 2019 product was to seek information on the plume location rather than getting absolute concentrations right. Hence, an extensive summary on the individual components contributing to the retrieval error is superfluous to our goal.

We have removed our comment on the injection height estimate after further consideration.

The core element of the manuscript is the investigation of the SO2-to-SO4 reactions. The use of the biexponential fit and the division into gas-phase and aqueous-phase pathways seems promising on a first sight. Are there any references, where you base this method on? When exploring Figure 6a, the monoexponential and biexponential fits appear very similar and it is hard to justify why the biexponential fit performs better. How can you make sure that the two reaction pathways can directly be mapped within a biexponential fit? The scattering of the data point remains widely spread while the exponents derived from the biexponential fit are fairly close. Please re-evaluate this analysis and provide more evidence for the pathway assumption.

We thank the reviewer for raising their concerns with the biexponential fitting. A similar comment was made from another reviewer. In hindsight, we acknowledge that attempting to separate the in-plume SO$_2$ into its gaseous and aqueous pathways was too much of stretch for our dataset. We now only fit to an exponential with a single decay constant. The derived gaseous-phase and aqueous-phase oxidation rate constants are now replaced with a single value generalising them. However, we

believe that our efforts to explore the complexity of volcanic $SO_2$ oxidation is still worth mentioning and a suggested direction for future works has been included.

Minor comments:

Please review all citations (e.g., in line 50 Aas et al., 2015 is cited, but does not exist in the reference list).

Corrected, thank you.

Please check punctuation. Extra commas would increase readability. And e.g., line 209 misses a "." after "respectively".

Corrected, thank you.

Regarding the tables' captions, these are typically written above the tables. Please revise.

Corrected, thank you.

Line 36: "ugm-3" must probably be µgm-3

Corrected, thank you.

Line 55: delete doubled "aerosol"

Corrected, thank you.

Line 111: SO2 column load and plume height "are" derived…

Corrected, thank you.

In line 138, the calculation of monthly surface mass concentration climatology is not fully clear. What time span is used here? Is the full temporal coverage mentioned in the text corresponding variable for the different stations and corresponds to the column "Temporal coverage" in table 1?

Yes, you are correct. We have amended this explanation within the manuscript to provide added clarity.

Table 1: I just count 22 EMEP stations being listed in the table.

We find 25. Perhaps part of the table has not rendered probably during upload as it spans multiple pages.

Figure 2: Why mentioning the 21 UTC sampling in the caption, if the figure shows simulation results in the morning?

Corrected, thank you.

Lines 264-265: Here, sharp peaks and troughs are mentioned but probably only the troughs are discussed. This is confusing. Please also check the dates listed here. These are not well recognisable in Fig. 3c.

This statement has been removed as is no longer required and the dates have been checked.

Figure 4: Please define "pollution event". What is the timeframe of high sulphur concentrations for such an event? And can events occur multiple times a day?

"Pollution event" is defined in Sect 2.2. and we have added a comment to the figure caption to clarify this. No, only one event per day is considered here.

In lines 337, 387, and 401, there are 22 EMEP stations mentioned. However, the explanation before states 20 stations. Please check!

The change in the number of stations is because we are discussing different totals in these sections. To clarify, this study uses data from 25 EMEP stations. Of the 25 EMEP stations, 22 experience at least one pollution event, regardless whether it is Holuhraun attributed or not, during September and October 2014. Then, of the 22 EMEP stations experiencing a pollution event/s, 19 experience at least one event that has been attributed to Holuhraun.

Or put simply, the station counts change as follows:

- 25 -> 22 as 3 EMEP stations did not experience a pollution event of *any* origin
- 22 -> 19 as 3 EMEP stations did not experience a pollution event of Holuhraun origin

We have added further clarity within the manuscript to address this confusion.

Line 371: Meaning of "… as the models are essentially trying to correctly capture the behaviour of two pollutants as opposed to one" is unclear. Please rephrase.

Our statement was flawed and has been removed.

Line 425: Please add SO2-to-SO4 again before "ratio of 31+-4".

Corrected, thank you.

Table 5: Should ECHAM6.w-HAM2.3 have a footnote indicated by the "*"? If yes, where is the explanation?

This was used to identify that this model's output could not be fitted successfully to a biexponential fit. Since we no longer do this fitting, the "*" has been dropped.

Line 462: A comparison against IASI SO2 retrievals "shows" that…

Corrected, thank you.

Lines 464-465: Please be more precise here. What is an underestimation of a distribution?

Corrected, thank you.

Line 473: "whilst considering everything else equal" Is this really the case? What is about the different resolutions, different chemical mechanisms, different transport schemes? Please extend this discussion.

We have reworded our comment for clarity. The intent was to state the importance of understanding the volcanic perturbation to the region before exploring the impacts this perturbation has on aerosol-cloud interactions, rather than to discuss the model differences.

Carboni, E., Mather, T. A., Schmidt, A., Grainger, R. G., Pfeffer, M. A., Ialongo, I., and Theys, N.: Satellite-derived sulfur dioxide (SO2) emissions from the 2014–2015 Holuhraun eruption (Iceland), Atmos. Chem. Phys., 19, 4851–4862, doi:10.5194/acp-19-4851-2019, 2019.

Walker, J. C., Dudhia, A., and Carboni, E.: An effective method for the detection of trace species demonstrated using the MetOp Infrared Atmospheric Sounding Interferometer, Atmos. Meas. Tech., 4, 1567–1580, doi:10.5194/amt-4-1567-2011, 2011.

Walker, J. C., Carboni, E., Dudhia, A., and Grainger, R. G.: Improved detection of sulphur dioxide in volcanic plumes using satellite-based hyperspectral infrared measurements: Application to the Eyjafjallajökull 2010 eruption, J. Geophys. Res., 117, D00U16, doi:10.1029/2011JD016810, 2012.

---

## Author Response (AR3)

**Referee Comment**

The trajectory analysis is somewhat problematic. First of all, how are the 27 members of the trajectory ensembles (line 174) different from each other? This is not explained in the text. Second, all EMEP stations are located in the atmospheric boundary layer, where air mass trajectories are not well representing the properties of the flow, due to turbulence. This will likely affect the quality of the attribution of events to Holuhraun (or not). Third, the definition of "vicinity" of the Holuhraun eruption is highly subjective. Depending, e.g., on the transport time and distance, trajectory errors will likely be very much case-dependent, and a single "vicinity area" might not be appropriate for all cases (e.g., stations closer to Holuhraun will have a greater chance of hitting the defined vicinity area.

**Author's Response**

Thank you for raising your concerns regarding the trajectory analysis. We hope the following changes reassure you that our methodology is sound.

Firstly, a more detailed description on the trajectory analysis has been provided. This includes a clarification on how the 27 ensemble members differ due to small perturbations in the input meteorology data (i.e. offset by a fixed grid factor, a maximum of 1.0° of latitude/longitude in the horizontal and 0.01 sigma units in the vertical, and so all possible offsets result in the 27 members).

Secondly, we acknowledge there is often greater uncertainty in using single-particle trajectories as opposed to dispersion modelling. Nevertheless, single-particle trajectory frameworks like ours have been widely used in many previous studies to characterise long-range transport (e.g. Nieminen et al., 2015; Räty at al., 2023; Väänänen et al., 2013). Plus, our goal is to inter-compare models consistently rather than through a rigorous dispersion exercise, and so believe our trajectory framework is sufficient to achieve this. We have added additional comments to the manuscript discussing this.

In addition, the reviewer is right in that these uncertainties may affect the quality of attribution. However, the Holuhraun emissions were so substantial within the region that it would be extremely unlikely that most identified events were misattributed to Holuhraun. Moreover, sulphurous surface concentrations are rather low over Europe in recent times and spikes in $SO_2$ are a rare occurrence.

Finally, we agree that the "vicinity" of the Holuhraun eruption is subjective. To address this, we have taken a more quantitative approach. We now define multiple spherical bounding areas with radii increasing with a station's increasing distance from Holuhraun. These radii values are based on the positional error of a trajectory being approximately 10–30% of the total distance travelled (Stohl, 1998). A special case is made for the Irafoss station due to the typical spatial resolution of the trajectories being greater than the error estimated from 10-30% of the distance. Consequently, we define the bounding radius here using local wind speeds following the methodology set out in Hughes et al. 2012.

**Author's Changes in Manuscript**

Lines 188 – 243.

Figure 1.

**Referee Comment**

The comparison between models and IASI data is not fully convincing. It seems model output is shown irrespective of whether IASI retrievals are available for a location or not. IASI retrievals can easily miss volcanic $SO_2$, e.g., underneath clouds. Thus, models should only be sampled in pixels where IASI $SO_2$ retrievals are actually made. The authors write that models often have larger plume areas than the IASI retrievals, which can be attributed to clouds affecting IASI. Still, it appears that many models actually have often smaller plume areas than IASI. This would even be worse when cloud screening is applied.

**Author's Response**

We have listened to your suggestion and agree the better approach is to only sample models where successful IASI $SO_2$ retrievals have been made. Subsequently, our comparison with IASI retrievals is now only made on grid cells within the observed plume extent. On redoing the analysis, the modelled $SO_2$ plume heights now overestimate observations whilst no substantial changes in the $SO2$ mass burden comparison is found.

On the comment "many models actually have often smaller plume areas than IASI", we acknowledge that this is true across certain periods, yet overall we observe that the modelled plume extents are larger than those observed, particularly in October. We have improved Fig. 2 and the animation by explicitly distinguishing between regions inside and outside the observed plume extent in the model simulations (coloured vs hatched areas) to improve the evidence for this statement.

**Author's Changes in Manuscript**

Lines 263 – 376; 525 – 538.

Figures 2, 3, and 4.

Table 3.

**Referee Comment**

Figure 5: Since the conversion rate of $SO_2$ to sulfate is shown to be uncertain, I am wondering why Figure 5 does not also show a comparison for total sulphur ($SO_2$ + sulfate). This should provide the most robust comparison between the models and the observations.

**Author's Response**

A total sulphur comparison has been added to the figure and the discussion extended.

**Author's Changes in Manuscript**

Lines 429 – 438.

Figure 6.

**Referee Comment**

Figure 6: This is the core result of the paper and quite interesting. However, I am not at all convinced that the bi-exponential fit is any better than the mono-exponential fit. That the bi-exponential fit is better (line 417) is a trivial result. But is it really SIGNIFICANTLY better? The two e-folding times obtained are interpreted as gas-phase and aqueous-phase e-folding times. But I am concerned that the fit is not stable enough to reliably distinguish between the two. Furthermore, how do you know which e-folding time is which? The data per se do not give any information on the two processes, but the authors immediately jump to the conclusion that these are gas- and aqueous phase e-folding times. What is the evidence for this?

**Author's Response**

We thank the reviewer for raising their concerns with the biexponential fitting. A similar comment was made from another reviewer. In hindsight, we acknowledge that attempting to separate the in-plume $SO_2$ into its gaseous and aqueous pathways was too much of stretch for our dataset. We now only fit to an exponential with a single decay constant. The derived gaseous-phase and aqueous-phase oxidation rate constants are now replaced with a single value generalising them. However, we believe that our efforts to explore the complexity of volcanic $SO_2$ oxidation is still worth mentioning and a suggested direction for future works has been included.

**Author's Changes in Manuscript**

Lines 475 – 515; 562 – 572.

Figure 7.

Table 4.

**Referee Comment**

Figure 6: The aqueous-phase oxidation occurs only in clouds, so is a single e-folding time even appropriate to characterize this oxidation? This must be highly variable, depending on the time the $SO_2$ spends in a cloud.

**Author's Response**

See previous response.

**Author's Changes in Manuscript**

See previous comment on changes made.

**Referee Comment**

Figure 6: All events are exclusively attributed to Holuhraun. However, there are likely always (perhaps minor) contributions from other sources. How might these affect the results, especially far away from the volcano, where $SO_2$/sulfate ratios are low and even relatively small anthropogenic $SO_2$ emissions could affect the ratio substantially.

**Author's Response**

Our fitting considers both the uncertainty in the ratio and plume age, with the latter being by far the dominant source. Generally, the plume age error increases with increasing plume age. Consequently, ratios sampled in a mature plume have less influence on the fitting parameters than those sampled in a young plume. As you say, the ratios "far away from the volcano" are low and so possibly are affected by small anthropogenic emissions, whilst the ratios sampled closer to the eruption are larger and less likely to be significantly affected. However, these "far away" ratios have a limited influence on the fitting due to their associated large plume age errors, whereas the ratios with a low likelihood of anthropogenic impact have small plume age errors. Hence, the ratios that could be potentially affected by small anthropogenic $SO_2$ emissions substantially have a relatively minor influence on the overall oxidation rate constant and near-vent ratios derived.

In addition, a major goal of the EMEP network is to gather observations at locations where significant sources of local pollution are minimised (Tørseth et al., 2012). As such, anthropogenic $SO_2$ contributions in the observations used here should be minimal.

**Author's Changes in Manuscript**

Lines 143 – 146; 505 – 507.

**Referee Comment**

Line 435: A modelled event is considered successful if both $SO_2$ and sulfate concentrations are within a factor 5 of the observations. Doesn't this introduce a bias in the analysis? You show that modelled oxidation rates are too slow – in this case one would expect the model to often substantially overestimate observed $SO_2$ concentrations. But large overestimations would be substantially removed from the analysis, which would lead to biased results.

**Author's Response**

This is a good point and is seemingly a flaw in our analysis. As such, we now no longer only fit model output to the events captured within a factor of 5. All models are fitted to all observed events attributed to Holuhraun preventing this bias from impacting our results. The new analysis without the bias does not change remarkably.

**Author's Changes in Manuscript**

Lines 502 – 515.

Figure 7.

Table 4.

**Referee Comment**

Line 161 and Table 2: Why are ERA-Interim reanalyses used? These are superseded since quite a few years already by ERA5 reanalyses with better resolution, and which should have better quality!

**Author's Response**

The experiment was initialised prior to the public release of ERA5 (2019) which is why it is not included. As to why it has taken so long to reach the results stage, that is an unfortunate multitude of disruptions including COVID, a change of experiment lead, and submission errors.

**Author's Changes in Manuscript**

No changes.

**Referee Comment**

Lines 279-280: How do you know that varying IASI $SO_2$ burdens are due to changing IASI retrieval coverage and plumes passing in and out of the region, and not due to variations in emission flux? I don't think there is good enough data to prove that the emission flux was really constant.

**Author's Response**

We agree that there is not sufficient data to suggest the volcanic $SO_2$ emission flux was constant during September and October 2014 and that the coverage of IASI retrievals was too scarce to establish definitive conclusions. Rather, it is more likely variation existed (e.g. Thordarson and Hartley, 2015) and so contributes to the mass burden variation shown. Our comment has been amended to reflect this.

**Author's Changes in Manuscript**

Line 359 – 361.

**Referee Comment**

Lines 370-371: Why is a poor performance of concentration ratios expected? The two species are not simulated independently, so a plume in one species should always correlate with a plume in the other species.

**Author's Response**

Agreed, we have removed this comment.

**Author's Changes in Manuscript**

Lines 447 – 453.

**Referee Comment**

Table 3: Why does OsloCTM3 not have the data required for filling Table 3? This should be basic model output ($SO_2$ concentration fields) that is also needed for all other analyses?

**Author's Response**

The $SO_2$ columns loads and mass burdens for OsloCTM3 have now been included and the discussion amended accordingly. Unfortunately, from the diagnostics made available in the submission for this experiment it is not possible to derive an $SO_2$ plume height for OsloCTM3.

**Author's Changes in Manuscript**

Lines 363 – 376.

Figure 4.

Table 3.

**Referee Comment**

Line 55: word aerosol is duplicated in text

**Author's Response**

Corrected, thank you.

**Author's Changes in Manuscript**

Line 56.

**Referee Comment**

A multi-model intercomparison necessarily needs to consider differences in the models' characteristics as model physics/dynamics/chemistry and the model setup as parameterisations and model resolution to allow for a fair comparison and meaningful discussion. Therefore, I suggest adding the advection schemes used in Table 2. How can the OsloCTM3 not have a chemistry/aerosol module as a CTM? Furthermore, the vertical model level distribution is essential with respect to the volcanic emission plume. Please include the model layer thickness of the lowest model level and the number of model layers between 0 km and 3 km height in section 2.3 and consider this in the discussions! How do the different models differentiate with respect to vertical layering, vertical distribution of emissions etc.? Further, it is very hard to assess the performance of the different models individually, as the evaluation and the figures/tables often include just a selection of the models. It seems to be not a full and fair comparison between all models. Figure 2 does not include the CTM due to missing required diagnostics (lines 290-291). What does this mean? Why is it then being listed in table 3? Figure 3 just evaluates the performance of 3 models. Why are the model outputs not designed in the way that they are comparable? Please consider producing comparable model output to fully discuss all model performances.

**Author's Response**

Thank you for your suggestions on how to improve the model inter-comparison part of our study.

Firstly, you are correct in that the model vertical resolutions need to be considered more. Subsequently, details on the model vertical resolution have been added to Table 2 and considered during the discussions. However, we have not included the advection schemes. As the models are nudged to ERA-Interim reanalyses data, the dispersion is already constrained which was done, partly, to mitigate the transport errors. Hence, we feel added details on advection schemes is superfluous.

Secondly, you are right that OsloCTM3 has a chemistry scheme, and its omission is an error that has been corrected.

Moreover, all figures inter-comparing model output now contain all models where possible.

Finally, the $SO_2$ columns loads and mass burdens for OsloCTM3 have now been included and the discussion amended accordingly. Unfortunately, from the diagnostics made available in the submission for this experiment, it is only possible to derive monthly $SO_2$ plume height estimates for MIROC6.1-SPRINTARS and ECHAM6.3-HAM2.3-P3, whilst no estimate is possible for OsloCTM3. Whilst we agree with the reviewer that having all the diagnostics requested for the experiment would be ideal, the main aim of the experiment is the investigation of the aerosol-cloud interactions (ACIs) in Part 2. As a significant number of diagnostics were requested and the submissions made "pro-bono", some modelling centres focused their resources on the diagnostics required for the ACI study leaving some diagnostics absent for Part 1. We agree this limits the plume height inter-comparison, yet still believe the discussions on the output available to the experiment warrants inclusion.

**Author's Changes in Manuscript**

Lines 34 – 35; 363 – 376; 514 – 515; 547 – 549; 559 – 560.

Table 2.

Figures 4, 6, and 7.

**Referee Comment**

The setup of the backward trajectories simulated with HYSPLIT needs clarification (lines 174-178). It remains unclear how the ensemble trajectories are designed. Are there 27 ensemble members defined for each station and each hour? If so, how are the ensemble members at the individual stations perturbed? Or do you have one ensemble member per hour for each station? However, 27 stations do not agree with table 1. Table 1 lists the starting heights for the trajectories at each station. How are these heights defined? Is there a basis for these heights as the station height above the surface height of the corresponding GCM's grid cell? How do you ensure that the trajectories allow for a fair comparison with ground-based observations? Further, are 2 % of the trajectories passing through the 3D bounding box (line 184) significant enough to connect these to the Holuhraun event?

**Author's Response**

Thank you for raising your concerns regarding the trajectory analysis. We hope the following changes reassure you that our methodology is sound.

Firstly, a more detailed description on the trajectory analysis has been provided. This includes a clarification on how the 27 ensemble members differ due to small perturbations in the input meteorology data (i.e. offset by a fixed grid factor, a maximum of 1.0° of latitude/longitude in the horizontal and 0.01 sigma units in the vertical, and so all possible offsets result in the 27 members).

Secondly, the trajectory starting heights for each station used offsets to the model surface to address differences between the orography in the driving meteorological data and reality.

Thirdly, we agree that the 2% threshold was too small to be confident in attributing a pollution event to Holuhraun emissions. The threshold has since been increased to 25%.

**Author's Changes in Manuscript**

Lines 188 – 243.

Figure 1.

**Referee Comment**

The assessment of the transport time (lines 185-192) is not fully clear and needs some rephrasing. I would have expected a circular influence region around the eruption site instead of a squared

bounded domain. Please also elaborate on the definition of the idealised trajectory points in Fig. 1. Do these relate to full hours? And how many black circles are attributed to which trajectory?

**Author's Response**

We agree that a circular influence region is more suitable and have made changes to our trajectory framework to include such. In our revised framework, the idealised trajectory points in Fig. 1 are no longer relevant and have been removed.

**Author's Changes in Manuscript**

Lines 207 – 243.

Figure 1.

**Referee Comment**

Regarding the IASI SO$_2$ retrieval, it remains unclear what "the SO$_2$ detection is positive" means (lines 112-114). Is 0.49 DU a detection limit? Or is it an individually defined value to discriminate volcanic SO$_2$ from other SO$_2$ (which is the climatology)? Please clarify this. Furthermore, in line 118, you are referring to meteorological temperature profiles for the height conversion. Where do these profiles come from? Are they standard profiles or from meteorological model analyses? To better understand the uncertainty of the IASI retrievals, it would be desirable to have a short summary of the different components contributing to the retrieval error. For example, do the uncertainties of assuming Gaussian profiles and the uncertainties of the temperature profiles contribute to the retrieval error? With respect to the discussion in line 249, please justify why the central height of a Gaussian SO$_2$ vertical profile can be an estimate for the injection height. There are enormous amounts of mass being distributed above the central height of a Gaussian profile.

**Author's Response**

Thank you for seeking additional clarity on the IASI SO$_2$ retrievals.

The 0.49 DU is a threshold set by Carboni et al., 2019 within their IASI retrieval algorithm and is chosen specifically for the Holuhraun eruption. The threshold is used in the detection scheme which is a linear retrieval with the SO$_2$ column load as the free parameter. The linear retrieval assumes the background SO$_2$ concentration follows a Gaussian distribution. The threshold is set substantially higher than the standard deviation of this distribution meaning a positive result is exceedingly likely to be significantly different to the background. We have improved the clarity of this section within the manuscript. Further details on the detection scheme can be found in Walker et al. 2011, 2012.

In terms of the height conversion, this is performed within the algorithm in Carboni et al., 2019 and is not a post processing step by us. They use atmospheric profiles from European Centre for Medium-Range Weather Forecasts (ECMWF) meteorological data.

We have added extra details on the IASI error estimates and provided references to seek further information. Although considered important, the reason for using the Carboni et al., 2019 product

was to seek information on the plume location rather than getting absolute concentrations right. Hence, an extensive summary on the individual components contributing to the retrieval error is superfluous to our goal.

We have removed our comment on the injection height estimate after further consideration.

**Author's Changes in Manuscript**

Lines 103 – 141.

**Referee Comment**

The core element of the manuscript is the investigation of the $SO_2$-to-$SO_4$ reactions. The use of the biexponential fit and the division into gas-phase and aqueous-phase pathways seems promising on a first sight. Are there any references, where you base this method on? When exploring Figure 6a, the monoexponential and biexponential fits appear very similar and it is hard to justify why the biexponential fit performs better. How can you make sure that the two reaction pathways can directly be mapped within a biexponential fit? The scattering of the data point remains widely spread while the exponents derived from the biexponential fit are fairly close. Please re-evaluate this analysis and provide more evidence for the pathway assumption.

**Author's Response**

We thank the reviewer for raising their concerns with the biexponential fitting. A similar comment was made from another reviewer. In hindsight, we acknowledge that attempting to separate the in-plume $SO_2$ into its gaseous and aqueous pathways was too much of stretch for our dataset. We now only fit to an exponential with a single decay constant. The derived gaseous-phase and aqueous-phase oxidation rate constants are now replaced with a single value generalising them. However, we believe that our efforts to explore the complexity of volcanic $SO_2$ oxidation is still worth mentioning and a suggested direction for future works has been included.

**Author's Changes in Manuscript**

Lines 475 – 515; 562 – 572.

Figure 7.

Table 4.

**Referee Comment**

Please review all citations (e.g., in line 50 Aas et al., 2015 is cited, but does not exist in the reference list).

**Author's Response**

Corrected, thank you.

**Author's Changes in Manuscript**

Lines 608 – 879.

**Referee Comment**

Please check punctuation. Extra commas would increase readability. And e.g., line 209 misses a "." after "respectively".

**Author's Response**

Corrected, thank you.

**Author's Changes in Manuscript**

Various improvements made throughout the manuscript.

**Referee Comment**

Regarding the tables' captions, these are typically written above the tables. Please revise.

**Author's Response**

Corrected, thank you.

**Author's Changes in Manuscript**

Tables 1, 2, 3, and 4.

**Referee Comment**

Line 36: "ugm-3" must probably be µgm-3.

**Author's Response**

Corrected, thank you.

**Referee Comment**

Line 55: delete doubled "aerosol"

**Author's Response**

Corrected, thank you.

**Author's Changes in Manuscript**

Line 56.

**Referee Comment**

Line 111: $SO_2$ column load and plume height "are" derived…

**Author's Response**

Corrected, thank you.

**Author's Changes in Manuscript**

Line 114.

**Referee Comment**

In line 138, the calculation of monthly surface mass concentration climatology is not fully clear. What time span is used here? Is the full temporal coverage mentioned in the text corresponding variable for the different stations and corresponds to the column "Temporal coverage" in table 1?

**Author's Response**

Yes, you are correct. We have amended this explanation within the manuscript to provide added clarity.

**Author's Changes in Manuscript**

Lines 154 – 158.

**Referee Comment**

Table 1: I just count 22 EMEP stations being listed in the table.

**Author's Response**

We find 25. Perhaps part of the table has not rendered probably during upload as it spans multiple pages.

**Author's Changes in Manuscript**

No changes.

**Referee Comment**

Figure 2: Why mentioning the 21 UTC sampling in the caption, if the figure shows simulation results in the morning?

**Author's Response**

Corrected, thank you.

**Author's Changes in Manuscript**

Figure 2.

**Referee Comment**

Lines 264-265: Here, sharp peaks and troughs are mentioned but probably only the troughs are discussed. This is confusing. Please also check the dates listed here. These are not well recognisable in Fig. 3c.

**Author's Response**

This statement has been removed as is no longer required and the dates have been checked.

**Author's Changes in Manuscript**

Lines 308 – 337.

**Referee Comment**

Figure 4: Please define "pollution event". What is the timeframe of high sulphur concentrations for such an event? And can events occur multiple times a day?

**Author's Response**

"Pollution event" is defined in Sect 2.2. and we have added a comment to the figure caption to clarify this. No, only one event per day is considered here.

**Author's Changes in Manuscript**

Figure 5.

**Referee Comment**

In lines 337, 387, and 401, there are 22 EMEP stations mentioned. However, the explanation before states 20 stations. Please check!

**Author's Response**

The change in the number of stations is because we are discussing different totals in these sections. To clarify, this study uses data from 25 EMEP stations. Of the 25 EMEP stations, 22 experience at least one pollution event, regardless whether it is Holuhraun attributed or not, during September and October 2014. Then, of the 22 EMEP stations experiencing a pollution event/s, 19 experience at least one event that has been attributed to Holuhraun.

Or put simply, the station counts change as follows:

- 25 -> 22 as 3 EMEP stations did not experience a pollution event of *any* origin
- 22 -> 19 as 3 EMEP stations did not experience a pollution event of Holuhraun origin

We have added further clarity within the manuscript to address this confusion.

**Author's Changes in Manuscript**

Lines 392 – 421.

**Referee Comment**

Line 371: Meaning of "… as the models are essentially trying to correctly capture the behaviour of two pollutants as opposed to one" is unclear. Please rephrase.

**Author's Response**

Our statement was flawed and has been removed.

**Author's Changes in Manuscript**

Lines 447 – 453.

**Referee Comment**

Line 425: Please add $SO_2$-to-$SO_4$ again before "ratio of 31+-4".

**Author's Response**

Corrected, thank you.

**Author's Changes in Manuscript**

Lines 475 – 492.

**Referee Comment**

Table 5: Should ECHAM6.w-HAM2.3 have a footnote indicated by the "*"? If yes, where is the explanation?

**Author's Response**

This was used to identify that this model's output could not be fitted successfully to a biexponential fit. Since we no longer do this fitting, the "*" has been dropped.

**Author's Changes in Manuscript**

Table 4.

**Referee Comment**

Line 462: A comparison against IASI $SO_2$ retrievals "shows" that…

**Author's Response**

Corrected, thank you.

**Author's Changes in Manuscript**

Line 527.

**Referee Comment**

Lines 464-465: Please be more precise here. What is an underestimation of a distribution?

**Author's Response**

Corrected, thank you.

**Author's Changes in Manuscript**

Lines 527 – 538.

**Referee Comment**

Line 473: "whilst considering everything else equal" Is this really the case? What is about the different resolutions, different chemical mechanisms, different transport schemes? Please extend this discussion.

**Author's Response**

We have reworded our comment for clarity. The intent was to state the importance of understanding the volcanic perturbation to the region before exploring the impacts this perturbation has on aerosol-cloud interactions, rather than to discuss the model differences.

**Author's Changes in Manuscript**

Lines 536 – 538.

**Referee Comment**

Please add a short explanation on how you derived the plume age that is visualized in Fig. 6 and 7, and firstly mentioned in line 429.

**Author's Response**

A short explanation on the plume age derivation is provided in the methodology (Sect. 2.4, lines: 230-233). We now refer the reader to this section explicitly in text. Also, we have added a reference to this section when discussing the plume's travel distance following Fig. 7.

**Author's Changes in Manuscript**

Lines: 428; 472

**Referee Comment**

In Fig. 7f, the exponential decay fit seems incorrect, because it does not align with the other panels. Please verify and adjust the discussion with respect to MIROC6.1-SPRINTARS accordingly.

**Author's Response**

The exponential decay is fitted using the same method applied to the observations and other models shown in Fig. 7. We acknowledge the difference between MIROC6.1-SPRINTARS and the other panels in text (lines: 498-499). Our reasoning for the poor fit in MIROC6.1-SPRINTARS is given in the manuscript as a combination of the following:

- Ratios sampled in the younger plume have a smaller total error, and so larger influence on the fitting, as opposed to ratios sampled in the mature plume (lines: 500-502).
- MIROC6.1-SPRINTARS underestimates the ratios in the young plume (30-60 h) (lines: 447-448; 507-508).

Hence, as these underestimated ratios in the young plume have relatively small total errors, the fitting procedure weights them more heavily than mature plume ratios which results in the sharp decay fit seen in Fig. 7f. We conclude the MIROC6.1-SPRINTARS discussion by stating that if the young plume ratios (30-60 h) were better captured by the model, a fit more in keeping with the other models and observations would likely be seen (lines: 506-509). We feel we address this seemingly incorrect fit sufficiently in the manuscript yet have amended the text to improve the clarity of our reasoning.

**Author's Changes in Manuscript**

Lines: 497-512

**Referee Comment**

Line 507-508: "The seemingly poor fit of MIROC6.1-SPRINTARS is likely due to the underestimation of the ratios between 24 h and 48 h (see Sect. 5)." The connection between the poor fit and the underestimation within 24-48 h remains unclear, as data is available for plume ages between ~5-210 h. Please clarify!

**Author's Response**

See response to previous comment.

**Author's Changes in Manuscript**

See changes to previous comment.

**Referee Comment**

Line 139: Are times in UTC or local time?

**Author's Response**

Times are in UTC.

**Author's Changes in Manuscript**

Line: 139

**Referee Comment**

Line 300: There is no "vertical profile" shown, only the plume extent. Please revise.

**Author's Response**

References to "vertical profile" have been removed and, where needed, replaced with "vertical extent".

**Author's Changes in Manuscript**

Lines: 301; 309; 331

**Referee Comment**

Fig. 4: Please remove panel (c) as it is the same as Fig.3(c). Also amend the in-text references. There is no need to show it twice.

**Author's Response**

Fig. 4c has been removed and the figure caption updated accordingly.

**Author's Changes in Manuscript**

Figure 4

Lines: 358

**Referee Comment**

Fig.6: Please add to the caption that the plume age is displayed.

**Author's Response**

Caption amended. Also, added a similar comment on the plume travel distance to the Fig. 7 caption.

**Author's Changes in Manuscript**

Lines: 423; 492

**Referee Comment**

Line 469: Missing "." at the end of the sentence.

**Author's Response**

Corrected, thank you.

**Author's Changes in Manuscript**

Line: 468

**Referee Comment**

Line 498: depicts -> depict

**Author's Response**

Corrected, thank you.

**Author's Changes in Manuscript**

Line: 497

**Referee Comment**

Line 509: capture -> captures

**Author's Response**

Corrected, thank you.

**Author's Changes in Manuscript**

Lines: 509-510

**Referee Comment**

Line 561: to the describe -> to describe

**Author's Response**

Corrected, thank you.

**Author's Changes in Manuscript**

Line: 562

**Referee Comment**

Line 574: help -> helps

**Author's Response**

Corrected, thank you.

**Author's Changes in Manuscript**

Line: 575